# Open-Det: An Efficient Learning Framework for Open-Ended Detection

Guiping Cao [1 2]  Tao Wang [1 2]  Wenjian Huang [1]  Xiangyuan Lan [2 3]  Jianguo Zhang [1 2 4]  Dongmei Jiang [2]

## Abstract

Open-Ended object Detection (OED) is a novel and challenging task that detects objects and generates their category names in a free-form manner, without requiring additional vocabularies during inference. However, the existing OED models, such as GenerateU, require large-scale datasets for training, suffer from slow convergence, and exhibit limited performance. To address these issues, we present a novel and efficient **Open-Det** framework, consisting of four collaborative parts. Specifically, Open-Det accelerates model training in both the bounding box and object name generation process by reconstructing the *Object Detector* and the *Object Name Generator*. To bridge the semantic gap between Vision and Language modalities, we propose a *Vision-Language Aligner* with V-to-L and L-to-V alignment mechanisms, incorporating with the *Prompts Distiller* to transfer knowledge from the VLM into VL-prompts, enabling accurate object name generation for the LLM. In addition, we design a Masked Alignment Loss to eliminate contradictory supervision and introduce a Joint Loss to enhance classification, resulting in more efficient training. Compared to GenerateU, Open-Det, using only **1.5%** of the training data (0.077M vs. 5.077M), **20.8%** of the training epochs (31 vs. 149), and fewer GPU resources (**4 V100** vs. 16 A100), achieves even higher performance (**+1.0%** in $AP_r$). The source codes are available at: https://github.com/Med-Process/Open-Det.

---
[1]Research Institute of Trustworthy Autonomous Systems and Department of Computer Science and Engineering, Southern University of Science and Technology, Shenzhen 518055, China. [2]Pengcheng Laboratory, Shenzhen, China. [3]Pazhou Laboratory (Huangpu). [4]Guangdong Provincial Key Laboratory of Brain-inspired Intelligent Computation, Department of Computer Science and Engineering, Southern University of Science and Technology, Shenzhen 518055, China. Correspondence to: Xiangyuan Lan <lanxy@pcl.ac.cn>, Jianguo Zhang <zhangjg@sustech.edu.cn>.

*Proceedings of the $42^{nd}$ International Conference on Machine Learning*, Vancouver, Canada. PMLR 267, 2025. Copyright 2025 by the author(s).

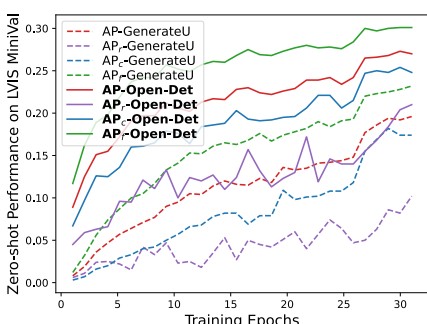

Figure 1: Performance curves of GenerateU and Open-Det, trained on the VG and evaluated on zero-shot LVIS MiniVal.

## 1. Introduction

Open Vocabulary object Detection (OVD) (Zareian et al., 2021a; Gu et al., 2021; Lin et al., 2022; Wu et al., 2023b; Cheng et al., 2024) is a fundamental task in computer vision that largely extends detection abilities from conventional closed-set to open-set detection, allowing the location and identification of objects beyond fixed categories of training data. However, OVD still depends on additional vocabularies as input to obtain detection results during inference. This reliance creates a necessary dependency on supplementary language knowledge priors, significantly restricting the model's detection capabilities in the open-world scenario.

Recently, Open-Ended object Detection (OED) (Lin et al., 2024) has emerged as a more general and practical object detection task that eliminates the need for predefined object categories during the inference stage. The end-to-end framework of GenerateU (Lin et al., 2024) formulates the OED as a generative problem, enabling dense object detection and associated name generation in a free-form manner.

Despite its open-ended nature, this model's detection capabilities face notable challenges: it requires large-scale datasets and substantial GPU resources (*e.g.*,16 A100 GPUs) for training, and it suffers from low training efficiency, slow convergence, and suboptimal detection performance. These challenges arise from three main issues: (1) **Semantic Gap:** This framework directly feeds vision queries from the object detector into the Large Language Model (LLM) to generate object names. Although it employs Vision-Language

alignment loss for these queries, the approach still faces challenges in effectively bridging the semantic gap between the *Vision* and *Language* modalities in high-dimensional feature space. This limitation negatively impacts both model training and the accuracy of object name generation. (2) **Contradictory Supervision:** The framework fails to consider the relationships between object categories within the image, resulting in contradictory supervisory losses and gradients that degrade training efficiency (the causes of Contradictory Alignment Loss are detailed in Sec. A.3). (3) **The Heavy-weight LLM Head and Noisy Alignment:** Generative LLMs typically operate with a large vocabulary size for text generation (*e.g.* 32,128 tokens in the T5 (Raffel et al., 2020) model, including a heavy-weight head of linear layer with 24.7M parameters), making training highly challenging. Additionally, during the early stages of training, vision queries are not well-aligned with language modalities, leading to noisy alignment. This sub-optimal alignment supervision can disrupt the pre-trained weights of the LLM Head, further reducing the training efficiency and slowing convergence, as detailed in Sec. 3.5.

Based on the analysis, a natural question arises: *Is there a more efficient OED framework that can accelerate training convergence, enhance training efficiency, and improve detection performance, while eliminating the reliance on large-scale datasets*?

To answer this question, we propose a novel and efficient Open-Ended Detection framework, termed **Open-Det**. As presented in Fig. 2, Open-Det consists of four key components: (1) Object Detector (**ODR**); (2) Prompts Distiller; (3) Object Name Generator; and (4) Vision-Language Aligner. Specifically, Open-Det enhances overall *convergence* speed in two key aspects: 1) **Accelerating the training for the box detector.** Inspired by the one-to-many matching approach to improve training convergence (Jia et al., 2023; Zong et al., 2023; Hu et al., 2023), we further design a decoder in *Object Detector* that incorporates a decoupled one-to-many and one-to-one matching structure. This design accelerates training without requiring an additional branch, reducing training complexity. 2)**Accelerating the training for the object name generation.** We enhance the training of the *Object Name Generator* in three ways: firstly, we freeze the heavy-weight *head* of the LLM (while keeping other weights active) in the early training stage and introduce a LoRa (Hu et al., 2021) head to accelerate training, which preserves the original pre-trained weights of the head while significantly reducing the trainable parameters of the LLM head; secondly, we design a Text Denoising training approach to facilitate the training and increase the robustness for the LLM; thirdly, we propose a Vision-to-Language Distillation Module (**VLD-M**) in *Prompts Distiller* to transfer the Vision-Language Alignment knowledge from the VLM into newly introduced **VL-prompts** for LLM. Instead

of directly using vision queries as input for the LLM in GenerateU, we utilize these VL-prompts, which effectively bridge the semantic gap between Vision and Language representations, to boost the training convergence of the LLM.

To further strengthen the alignment between Vision and Language modalities, we present a new Bidirectional Vision-Language Alignment module (**BVLA-M**) within the *Vision-Language Aligner*. This module is designed to improve the alignment of vision queries with text embeddings, thereby enhancing training efficiency and overall performance. Additionally, we introduce a novel **Masked Alignment Loss** to prevent the emergence of contradictory losses and their associated gradients, as detailed in Sec. 3.6 and Sec. A.3. We also present a **Joint Loss** for improving positive and negative queries classification (detailed in Sec. 3.6 and Sec. A.4). The main contributions are summarized as follows:

- We develop a novel and efficient end-to-end OED framework, referred to as **Open-Det**. It significantly accelerates training convergence in both the box *Detector* and the *Object Name Generator*, and it further distills the knowledge of Vision-Language alignment from a frozen VLM into VL-prompts using the proposed VLD-M, greatly enhancing the training for LLM.

- We improve training efficiency and performance by optimizing the alignment of Vision-Language modalities with BVLA-M, correcting contradictory supervision using *Masked Alignment Loss* and enhancing classification through a *Joint Loss* that associates IoU and alignment scores with binary classification scores.

- Open-Det outperforms **OVD** models like GLIP(A), achieving **+6.8%** in $AP_r$ and **+8.5%** in AP with only **11.7%** of training data. Additionally, it demonstrates significant superior efficiency compared to the **OED** model GenerateU: using just **1.5%** of training data, **20.8%** of training epochs, and fewer GPUs (4 V100 vs. 16 A100), it achieves a **+1.0%** improvement in $AP_r$.

## 2. Related Work

**Open-Vocabulary Object Detection (OVD).** Traditional closed-set object detection (Ren et al., 2015; He et al., 2017; Carion et al., 2020; Zhang et al., 2022a; Cao et al., 2024) (COD) can identify only fixed object categories, requiring additional bounding box annotations and costly model retraining when new categories are introduced. To address this issue, the OVD task is introduced in OVR-CNN (Zareian et al., 2021b). OVD enables the model to generalize to new object categories without the need for annotations by learning a rich vocabulary during pretraining on large image-caption datasets. Then, a series of methods related to the large-scale Vision-Language Model (VLM) like CLIP (Radford et al., 2021) are utilized to address the challenge of

limited training data and advance the performance of OVD, including distilling the knowledge from CLIP into detector of ViLD (Gu et al., 2021), learning continuous shared prompt representations of DetPro (Du et al., 2022), extending CLIP to learn region-level visual representations of RegionCLIP (Zhong et al., 2022), and forming bags of regions in BARON (Wu et al., 2023a). The transformer-based detector is also extended to the OVD task. MDETR (Kamath et al., 2021) utilizes a transformer-based architecture that allows for detecting objects conditioned on raw text queries. OV-DETR (Zang et al., 2022) achieves OVD through conditional matching. VL-DET (Lin et al., 2022) builds a unified framework to formulate object-language alignment as a set matching problem. CORA (Wu et al., 2023b) enhances the region-text distribution by introducing the region prompting and anchor pre-matching. Although these methods achieve advanced performance, they encounter common challenges related to region-text alignment and limited generalization ability in detecting new categories. Open World Object Detection (Joseph et al., 2021) introduces a framework that enables the detector to label unknown objects as "unknown" and incrementally learn these identified unknown categories without forgetting previously learned classes. However, this approach still requires new label priors for each incremental learning phase, restricting its practical applicability.

Recently, learning from a diverse range of data sources, such as image-text pairs and grounding data, has gained more attention in OVD to enhance the generalization ability of visual concepts. Detic (Zhou et al., 2022) employs a dual-branch approach for classification and box prediction, where the class branch is trained on a large-scale dataset to achieve OVD through image-level supervision. GLIP (Li et al., 2022b) unifies object detection and phrase grounding for pre-training with grounding data and image-text pairs. GLIPv2 (Zhang et al., 2022b) further streamlines the training pipeline and enhances the synergy between localization and understanding by integrating localization and Vision-Language pre-training. Based on GLIP, Grounding DINO (Liu et al., 2023) combines DINO (Zhang et al., 2022a) with grounded pre-training to effectively fuse language and vision modalities. DetCLIP (Yao et al., 2022) enhances visual-concept modeling through a dictionary-enriched framework for parallel OVD pre-training. Its successor, DetCLIPv2 (Yao et al., 2023), unifies detection, grounding, and image-text pair data under a hybrid supervision approach. The latest iteration, DetCLIPv3 (Yao et al., 2024), advances the architecture with a high-performance detector capable of excelling in OVD and generating hierarchical labels for detected objects. Although DetCLIPv3 possesses generative capabilities, its reliance on large-scale datasets (over 50M), complex multi-stage training processes, and substantial GPU resource requirements (32/64 V100) greatly limit its further development and applicability.

**Open-Ended Object Detection (OED).** LLM (Raffel et al., 2020; Touvron et al., 2023; Achiam et al., 2023) and multi-modal VLM models (Radford et al., 2021; Li et al., 2022a; 2023), trained on extensive datasets of text or image-text pairs, have demonstrated remarkable model capacity and generalization performance in classification tasks. Naturally, integrating these models with detection frameworks presents new opportunities for achieving *vocabulary-free* detection for arbitrary classes. GenerateU (Lin et al., 2024) first introduces the OED problem, proposing an end-to-end framework that reformulates object detection as a generative task incorporating an LLM. The framework integrates an object detector with an encoder-decoder-based generative LLM, using vision queries from the detector as inputs for the LLM to enable free-form object name generation. However, this approach struggles with slow training convergence, low efficiency, large GPU resource demands, and high training costs. Thus, efficiently leveraging existing LLMs to train OED models with limited detection data remains a critical challenge that needs to be further addressed.

## 3. Method

### 3.1. Main Architecture of the Open-Det Framework

As illustrated in Fig. 2, the Open-Det framework comprises four collaborative components. During the *training stage*, the input image $I \in \mathbb{R}^{H \times W \times 3}$ (where $H$ and $W$ denote height and width) and its corresponding object names $T$ are fed into the *Object Detector* and the VLM, respectively. This generates decoder queries $Q_d$ for bounding box prediction and text embeddings $T_e$ for Vision-Language feature alignment. To enhance this alignment, we propose a novel Bidirectional Vision-Language Alignment module (BVLA-M) within the *Vision-Language Aligner*. This module enhances alignment scores (detailed in Sec. 3.3) and generates alignment indices $VL_{align}$ for query-text matching, enabling supervision for the *Object Detector*.

To improve the object name generation, we introduce a novel *Prompts Distiller* into the OED framework, replacing the direct use of vision queries as input for LLM in GenerateU. The Prompts Distiller utilizes VLD-M , which takes backbone features $B$, encoder features $E$, queries $Q_d$, text embeddings $T_e$, and $VL_{align}$ as inputs, to transfer the Vision-Language alignment knowledge from the VLM (such as CLIP (Radford et al., 2021)) to decoder queries $Q_d$, producing VL-prompts ($P_{vl}$). These VL-prompts are then fed into the LLM (*e.g.*, T5 (Raffel et al., 2020)) to generate more accurate object names. Finally, bounding box predictions and binary classification scores are derived from the decoder queries $Q_d$ and the object names are generated by the generative LLM. During the *inference* phase, the *Vision-Language Aligner* is omitted, achieving *vocabulary-free* detection and effectively reducing inference complexity.

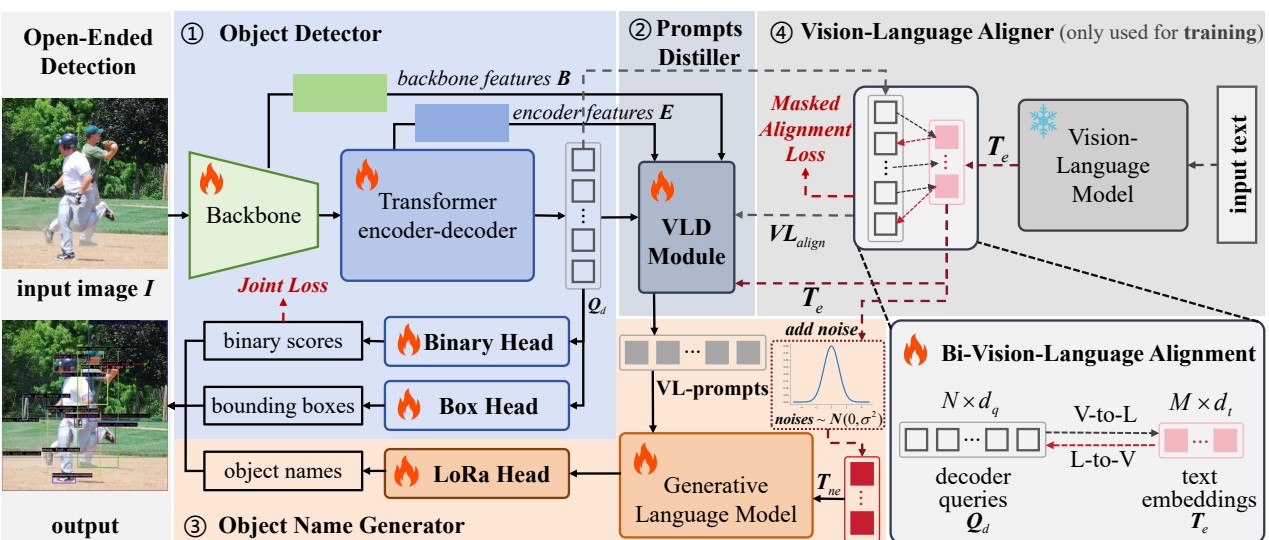

Figure 2: Main architecture of the Open-Det framework. It consists of 4 collaborative components: (1) **Object Detector (ODR)** for accelerating the bounding box training; (2) **Prompts Distiller** with Vision-to-Language Distillation module (**VLD-M**) to bridge the semantic gap between Vision and Language; (3) **Object Name Generator** with the Text Denoising approach to accelerate the training of the LoRa Head; (4) **Vision-Language Aligner** with **BVLA-M** to enhance the alignment of Vision and Language. The **Masked Alignment Loss** and **Joint Loss** are introduced for correcting the supervision information and enhancing binary classification consistency, respectively. Please refer to Sec. A.1 for simplified pipeline.

## 3.2. Object Detector

The *Object Detector* (ODR) plays a crucial role in generating bounding boxes and vision queries, which form the foundation for object name generation. Inspired by the efficiency of DINO (Zhang et al., 2022a), which enhances training through anchor box denoising, and the success of one-to-many matching in accelerating convergence (Jia et al., 2023; Zong et al., 2023; Hu et al., 2023), we further design a novel decoder structure, improving training convergence without requiring additional decoder branch and enabling flexible number of objects detection.

Unlike existing methods (Jia et al., 2023; Zong et al., 2023; Hu et al., 2023) that rely on additional branches or heads to combine one-to-one and one-to-many matching, we simplify the decoder design via decoupling these two matching approaches. Specifically, the first four layers employ one-to-many matching with cross-attention to enhance box localization, while the last two layers use one-to-one matching with self-attention to eliminate duplicate detections. Additionally, we aim to develop a more flexible OED framework capable of detecting a variable number of objects, to address the limitation of existing COD, VOD, and OED frameworks that are restricted to a fixed number of objects due to their reliance on predefined queries. To achieve this, we introduce a threshold-based query selection method that adaptively chooses queries $Q_d$ from encoder tokens. This design ensures both efficiency and flexibility in detecting varying

numbers of objects. The process can be formulated as:

$$Q_{id} = \{e_t \in E | \sigma(Linear(e_t)) > \lambda\} \qquad (1)$$

where $Q_{id}$ indexes each token $e_t$ from the encoder features $E$, and the decoder queries $Q_d$ are selected from $E$ according $Q_{id}$. The probability of each token $e_t$ being chosen as a query is calculated by a $Linear$ head layer, where $\sigma$ is the *sigmoid* function and $\lambda$ is the threshold (default: 0.05).

## 3.3. Vision-Language Aligner with BVLA-M

Vision-Language modalities alignment is a core component in the label-matching process. The alignment score is computed by measuring the similarity between the vision queries $Q_d$ and the text embeddings $T_e$. However, the channel dimension of $Q_d$ is usually smaller than that of $T_e$; for example, $Q_d$ has 256 channels while $T_e$ has 768. This significant dimensionality gap results in insufficient information in the vision queries compared to text embeddings. Consequently, directly aligning these two features by mapping $Q_d$ into a higher-dimensional space along the channel dimension, as done in existing methods (Gu et al., 2021; Lin et al., 2022; 2024), may lead to suboptimal Vision-Language alignment.

To address this issue, we propose a simple yet effective Bidirectional Vision-Language Alignment module (**BVLA-M**) to enhance the Vision-Language alignment supervision during training. Specifically, as shown in Fig. 2, we calculate the alignment scores from both Vision-to-Language (V-to-L)

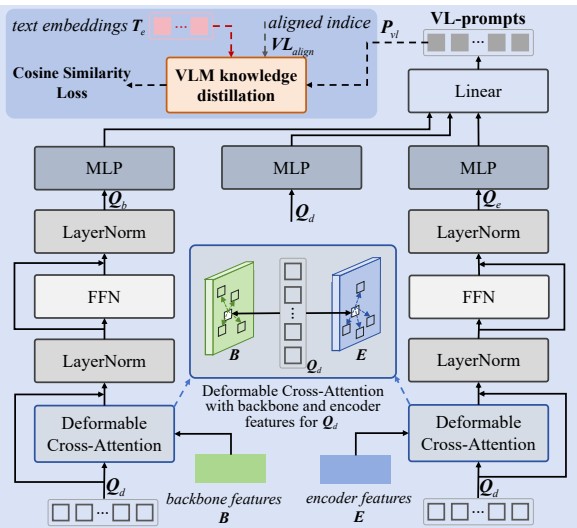

Figure 3: Overall architecture of the proposed VLD-M.

and Language-to-Vision (L-to-V) perspectives to strengthen the alignment between the Vision and Language modalities. This process can be formulated as:

$$S_{align} = cos(Q_d \times M_{VL}, T_e) + cos(Q_d, T_e \times M_{LV}) \quad (2)$$

where $Q_d \in \mathbb{R}^{N \times d_q}$ ($N$ is the number of queries and $d_q$ denotes the channel dimension), and $T_e \in \mathbb{R}^{M \times d_t}$ ($M$ is the number of text embeddings and $d_t$ is the channel dimension). $S_{align} \in \mathbb{R}^{N \times M}$ is the alignment score matrix, and $cos$ denotes the cosine similarity. $M_{VL}$ and $M_{LV}$ are the transformation matrices for V-to-L and L-to-V, respectively.

### 3.4. Vision-to-Language Distillation Module

The existing VLM models (Radford et al., 2021; Jia et al., 2021a) are usually trained with easily accessible web-scale *image-text* paired data. In contrast, the fine-grained *region-text* data is hard to obtain due to the challenges of complex labeling and its high cost (Zhong et al., 2022; Wu et al., 2023a;b). This difficulty poses a significant challenge for achieving effective region-text alignment in VLMs.

To address this issue, we propose a Vision-to-Language Distillation Module (**VLD-M**), as presented in Fig. 3. Existing methods like RegionCLIP (Zhong et al., 2022) generate additional pseudo-labels for region-text data using VLMs, while DVDet (Jin et al., 2024) transforms regional embeddings into image-like representations by cropping expanded predicted bounding boxes with handcrafted parameters to include background areas. In contrast, our VLD-M enhances queries by transforming regional embeddings $Q_d$ into *image-like* representations and further transferring knowledge from the VLM into these queries. This approach effectively bridges the gap between vision queries and image-level text embeddings from the VLM.

Specifically, VLD-M *adaptively* enriches the background information around $Q_d$ by interacting it with the backbone $B$ and encoder features $E$ through deformable cross-attention (Zhu et al., 2020) operations. This process adaptively samples token offsets around the query object, allowing it to learn relevant background information, as illustrated in Fig. 3. The feed-forward network (FFN) is used to re-weight queries, producing updated $Q_b$ and $Q_e$. These updated queries, along with the original $Q_d$, are then passed through MLP layers—comprising two linear layers and an activation layer—to project the lower-dimensional queries into the same dimensional space as $T_e$ along the channel dimension. Finally, a linear layer fuses these queries, generating the VL-prompts $P_{vl}$.

Knowledge distillation from the VLM into VL-prompts $P_{vl}$ is achieved using Cosine Similarity Loss as supervision to facilitate object name generation. Given the text embeddings $T_e$ and aligned indices $VL_{align}$ from the BVLA-M, we first select the VL-prompts and text embedding pairs based on these aligned indices and compute the Cosine Similarity Loss between $P_{vl}$ and $T_e$, enabling effective knowledge distillation from the VLM to the VL-prompts $P_{vl}$.

### 3.5. Object Name Generator

**Noisy Alignment.** In the early stages of training, the decoder queries $Q_d$ are under-trained and exhibit a large semantic gap compared to the text embeddings, leading to noisy and low-quality query-text alignment. Training with such noises risks disrupting the pre-trained weights of the LLM, especially the heavy-weight head layer responsible for text generation, leading to inefficient and noisy training.

To address this issue, we *freeze* the head of the LLM (while keeping other weights active) in early training stage and introduce a LoRa Head (Hu et al., 2021) to accelerate training (detailed in Sec.A.2). This approach prevents disruption of the pre-trained LLM head weights and significantly reduces the trainable parameters of the LLM head. Furthermore, we observe that when text embeddings $T_e$ are fed into the LLM (*e.g.*, T5 (Raffel et al., 2020)) for text reconstruction, the training loss decreases rapidly, indicating T5 model can effectively process these embeddings to reconstruct the text. Leveraging this insight, as shown in Fig. 2, we enhance the robustness of the T5 model by adding Gaussian noise into the text embeddings and feeding them into T5 for text reconstruction. The noise follows a normal distribution $N(0, \sigma^2)$, where $\sigma$ is the standard deviation of the text embeddings $T_e$. This *Text Denoising* approach stabilizes training and improves the model's ability to handle noisy inputs.

### 3.6. Masked Alignment Loss and Joint Loss

In GenerateU, the Vision-Language alignment loss (namely the BCE loss) is used to enhance the alignment between

queries and text embeddings. This loss strengthens the *similarity* between matched query-text pairs while applying negative constraints to increase the *dissimilarity* between the query and all unmatched text embeddings in the mini-batch. However, it fails to account for the relationships among object names within the images or mini-batch, leading to *contradictory losses* and *gradients* that ultimately reduce training efficiency, as detailed in Sec. A.3.

To address this issue, we introduce a Masked Alignment Loss (**MAL**), which generates a binary mask $M = T_e \times T_e^\top$ (where $M \in \mathbb{R}^{M \times M}$ and $M$ is the number of text embeddings) to compute the similarity between all paired text embeddings. The mask is binarized using a threshold $\tau$ (default: 0.99), assigning a value of 1 for the same categories and a value of 0 to others. The MAL can be formulated as:

$$\mathcal{L}_{\text{MAL}} = - \frac{1}{NM} [(\boldsymbol{VL}_{align} \times \boldsymbol{M}) \odot log(\boldsymbol{S}_{align}) + \\ (1 - \boldsymbol{VL}_{align} \times \boldsymbol{M}) \odot log(1 - \boldsymbol{S}_{align})] \quad (3)$$

where $N$ and $M$ are the number of queries and text embeddings, respectively. The matrix $\boldsymbol{VL}_{align} \in \mathbb{R}^{N \times M}$ contains the aligned Vision-Language indices in one-hot format, while $\boldsymbol{S}_{align} \in \mathbb{R}^{N \times M}$ represents the alignment score matrix. Our MAL loss works by utilizing this mask to calibrate the alignment labels, thereby preventing the occurrence of contradictory losses and gradients.

Additionally, our MAL approach differs fundamentally from ScaleDet (Chen et al., 2023) in both methodology and objectives: ScaleDet unifies text labels via semantic similarity (as soft label in MSE) to combine multi-dataset training; in contrast, MAL resolves query-text matching conflicts through similarity-binarized BCE updates.

**Joint Loss for Positive and Negative Object Predictions.** The OED framework detects objects and generates their names automatically during *inference*, meaning it cannot compute the Vision-Language alignment scores of detected objects for matching and selection of results. Additionally, the number of detected objects is typically large, necessitating deduplication. Therefore, an additional head is applied to perform binary classification on the decoder queries $\boldsymbol{Q}_d$, identifying and filtering out duplicate detections.

However, the binary labels for each query are dynamically assigned based on the Vision-Language alignment and matching process: matched queries are labeled as positive samples, while unmatched queries are labeled as negative samples. This *dynamic labeling* increases the difficulty of binary prediction for this head. In the Open-Det framework, similar to DETR-like models (Carion et al., 2020; Zhang et al., 2022a; Hu et al., 2023), the matching mechanism evaluates both the *IoU score* and the *alignment score* (analogous to the classification score in COD detectors) between predicted boxes and ground truth boxes. Queries with higher

IoU and alignment scores are more likely to match ground-truth boxes, while those with lower scores are more likely to remain unmatched. Based on this insight, we propose a novel **Joint Loss** to improve binary predictions by integrating the binary score with IoU and alignment scores to enhance their consistency. This loss can be formulated as:

$$\mathcal{L}_{\text{JL}} = - \frac{1}{N} \sum_{i=1}^{N} \left[ (\sqrt{p_i^\alpha s_i^\alpha u_i^{1-2\alpha}} - p_i)^2 y_i \log(p_i) + \\ p_i^2 (1 - \sqrt{p_i^\alpha s_i^\alpha u_i^{1-2\alpha}})(1 - y_i) \log(1 - p_i) \right] \quad (4)$$

where $y_i \in \{0, 1\}$ is the binary label, while $p_i$, $s_i$, and $u_i$ denote the binary score, alignment score, and IoU score, respectively. $N$ denotes the number of queries. $\alpha$ (default: 0.25) is the scale factor. The square root operation is applied to prevent the product of these three scores from becoming too small, which helps maintain numerical stability for training. The effectiveness analysis of Joint Loss, along with its differences from Focal loss (Lin et al., 2017) and BCE loss, is detailed in Sec. A.4 of the Appendix. The total losses for model training are detailed in Sec. A.5.

## 4. Experiments

**Datasets.** We train our model with a small set of detection data Visual Genome (VG) (Krishna et al., 2017), which contains 77,398 images for training. Following the pioneering work of GenerateU (Lin et al., 2024) in OED, our model is evaluated on the commonly used zero-shot LVIS (Gupta et al., 2019) dataset, which contains 1,203 categories. The COCO2017 (Lin et al., 2014) and Objects365 (Shao et al., 2019) are also used for performance evaluation.

**Evaluation Metrics.** Following the evaluation methodology of the GenerateU framework, we compute the similarity score between generated object names and annotated category names using a fixed pre-trained text encoder. The evaluation metrics include: average precision for rare categories ($AP_r$), common categories ($AP_c$), frequency categories ($AP_f$), and all categories ($AP$), respectively. To ensure a fair comparison, we adhere to the protocols established in popular works (Kamath et al., 2021; Li et al., 2022b; Lin et al., 2024), evaluating on the $5k$ MiniVal subset of the LVIS (Gupta et al., 2019) dataset.

**Implementation Details.** For a fair comparison, Open-Det utilizes the same backbone model (*e.g.*, Swin-Tiny and Swin-Large (Liu et al., 2021)) and FlanT5-base (Chung et al., 2024) generative language model as used in the GenerateU framework. Unless otherwise specified, our models are trained with a mini-batch size of 8 on 4 Tesla V100 GPUs, using the AdamW optimizer (Loshchilov, 2017) with a weight decay of 0.05. The learning rates are configured as

Table 1: Comparison results of the zero-shot domain transfer on LVIS MiniVal dataset. In row 2, LVIS* indicates that the model is initially trained on GoldG (Kamath et al., 2021) and then fine-tuned using 10% of the LVIS data in MDETR. The GRIT5M used for GenerateU training includes 5 million images, selected from the larger grounding dataset of GRIT (Peng et al., 2023), which contains a total of 90.6M images. The details about the training epoch are presented in Sec. B.1.

| Model | Backbone | Pre-Train Data | Data Size | Vocabulary-Free | Epochs | $AP_r$ | $AP_c$ | $AP_f$ | AP |
|---|---|---|---|---|---|---|---|---|---|
| MDETR (Kamath et al., 2021) | ResNet101 | GoldG,LVIS* | 0.812M | ✗ | - | 20.9 | 24.9 | 24.3 | 24.2 |
| MaskRCNN (He et al., 2017) | ResNet101 | LVIS | 0.118M | ✗ | - | 26.3 | 34.0 | 33.9 | 33.3 |
| Deformable DETR (Zhu et al., 2020) | Swin-Tiny | LVIS | 0.118M | ✗ | - | 24.2 | 36.0 | 38.2 | 36.0 |
| GLIP(A) (Li et al., 2022b) | Swin-Tiny | O365 | 0.660M | ✗ | - | 14.2 | 13.9 | 23.4 | 18.5 |
| GLIP(C) (Li et al., 2022b) | Swin-Tiny | O365,GoldG | 1.460M | ✗ | - | 17.7 | 19.5 | 31.0 | 24.9 |
| GLIP(C) (Li et al., 2022b) | Swin-Tiny | O365,GoldG,CAP4M | 5.456M | ✗ | - | 20.8 | 21.4 | 31.0 | 26.0 |
| Grounding-DINO (Liu et al., 2023) | Swin-Tiny | O365,GoldG | 1.460M | ✗ | - | 14.4 | 19.6 | 32.2 | 25.6 |
| Grounding-DINO (Liu et al., 2023) | Swin-Tiny | O365,GoldG,Cap4M | 5.460M | ✗ | - | 18.1 | 23.3 | 32.7 | 27.4 |
| GenerateU (Lin et al., 2024) | Swin-Tiny | VG | 0.077M | ✓ | 149 | 17.4 | 22.4 | 29.6 | 25.4 |
| GenerateU (Lin et al., 2024) | Swin-Tiny | VG,GRIT5M | 5.077M | ✓ | - | 20.0 | 24.9 | 29.8 | 26.8 |
| **Open-Det (ours)** | Swin-Tiny | VG | 0.077M | ✓ | **31** | $21.0_{\uparrow3.6}$ | $24.8_{\uparrow2.4}$ | $30.1_{\uparrow0.5}$ | $27.0_{\uparrow1.6}$ |
| **Open-Det (ours)** | Swin-Tiny | VG | 0.077M | ✓ | 50 | $\textbf{21.9}_{\uparrow4.5}$ | $\textbf{25.1}_{\uparrow2.7}$ | $\textbf{30.4}_{\uparrow0.8}$ | $\textbf{27.4}_{\uparrow2.0}$ |
| GenerateU (Lin et al., 2024) | Swin-Large | VG,GRIT5M | 5.077M | ✓ | - | 22.3 | 25.2 | 31.4 | 27.9 |
| **Open-Det (ours)** | Swin-Small | VG | 0.077M | ✓ | 31 | $26.0_{\uparrow3.7}$ | $28.6_{\uparrow3.4}$ | $32.8_{\uparrow1.4}$ | $30.4_{\uparrow2.5}$ |
| **Open-Det (ours)** | Swin-Large | VG | 0.077M | ✓ | 31 | $\textbf{31.2}_{\uparrow8.9}$ | $\textbf{32.1}_{\uparrow6.9}$ | $\textbf{34.3}_{\uparrow2.9}$ | $\textbf{33.1}_{\uparrow5.2}$ |

follows: $1 \times 10^{-4}$ for both the *Object Detector* and *Prompts Distiller*, and $2 \times 10^{-4}$ for the *Object Name Generator*.

### 4.1. Main Results

Compared to **OVD** models like GLIP (Li et al., 2022b), Open-Det demonstrates significant advantages by eliminating the need for additional vocabulary priors in inference and achieving higher model efficiency. As shown in Table 1, Open-Det achieves superior performance with substantially reduced training data. When trained using only 0.077M data (*11.7%* of the data used by GLIP(A)), Open-Det outperforms GLIP(A) by **+6.8%** in $AP_r$ and **+8.5%** in AP. Even compared to GLIP(C), which is trained on a much larger dataset of 5.456M, Open-Det still achieves a **+1.0%** higher AP while using just *1.4%* of the training data. Additionally, Open-Det's data efficiency is further validated against Grounding DINO (Liu et al., 2023), maintaining consistent advantages with fewer training resources.

Compared to the pioneering **OED** framework of GenerateU (Lin et al., 2024), Open-Det framework demonstrates faster convergence and higher performance. Specifically, Open-Det achieves significant performance improvements, with **+3.6%** in $AP_r$ and **+1.6%** in AP, while requiring only 20.8% of the training epochs compared to GenerateU. Furthermore, even when GenerateU is trained with additional grounding data from GRIT5M (Peng et al., 2023), Open-Det still outperforms it by **+1.0%** in $AP_r$, while using only *1.5%* of the training data. Fig 1 illustrates the performance curves in relation to the training epochs, clearly highlighting Open-Det's advantages in both performance and faster convergence speed. These results underscore Open-Det's efficiency, scalability, and effectiveness in the OED task.

Extending Open-Det's training from 31 to 50 epochs yielded a **+0.9%** improvement in $AP_r$ and **+0.4%** in AP. Notably, Open-Det trained solely on VG dataset outperforms GenerateU (trained on both VG and GRIT5M), demonstrating **+1.9%** higher $AP_r$ and **+0.6%** higher AP.

We further evaluate Open-Det with two larger backbones: Swin-Small (using 4 V100 GPUs) and Swin-Large (using 4 A800 GPUs). As shown in Table 1, using only 1.5% training data, Open-Det-Swin-S achieves an improvement of **+3.7%** in $AP_r$ (26.0% vs. 22.3%) and **+2.5%** in AP (30.4% vs. 27.9%) than GenerateU-Swin-Large, demonstrating its efficiency. When utilizing the larger backbone of Swin-Large, Open-Det-Swin-Large significantly outperforms GenerateU-Swin-Large by **+8.9%** in $AP_r$ (31.2% vs. 22.3%) and **+5.2%** in AP (33.1% vs. 27.9%), further confirming its superior effectiveness and efficiency.

**Evaluation on COCO and Objects 365.** Compared to OVD, OED can directly detect objects in novel data under zero-shot setting without requiring text priors. The results in Table 2 indicate that using less training data, Open-Det outperforms GenerateU by **+2.2%** AP on COCO2017 (Lin et al., 2014) and **+3.3%** AP on Objects365 (Shao et al., 2019), demonstrating both higher data efficiency and improved detection accuracy.

### 4.2. Ablation Studies

**Effectiveness of Open-Det Components.** In Table 3, the ablation results for each component in Open-Det demonstrate their contributions: (1) The proposed ODR and

Table 2: Evaluation on COCO and Objects 365 in a zero-shot setting. The evaluation metric is AP.

| Methods | Backbone | Pre-Train Data | COCO | Objects365 |
|---|---|---|---|---|
| GenerateU | Swin-Large | VG | 33.0 | 10.1 |
| GenerateU | Swin-Large | VG,GRIT | 33.6 | 10.5 |
| **Open-Det** | Swin-Large | VG | $35.8_{\uparrow2.2}$ | $13.8_{\uparrow3.3}$ |

BVLA-M improve the performance by enhancing both the box convergence and the alignment between Vision and Language modalities. (2) The VLD-M further boosts performance by transferring knowledge from the VLM to the VL-prompts, achieving notable gains of **+1.6%** in $AP_r$, **+3.9%** in $AP_c$, and **+2.8%** in AP. This result demonstrates its critical effectiveness in bridging the semantic gap between vision and language domains. (3) The Object Name Generator (ONG), incorporating the LoRa Head and Text Denoising training approach, plays a critical role in improving the rare class and overall performance. When combined with specially designed loss functions—Masked Alignment Loss and Joint Loss—Open-Det achieves significant improvements of **+4.1%** in $AP_r$ and **+0.7%** in AP. These results effectively confirm their roles in accelerating convergence, enhancing training efficiency, and improving performance.

Table 3: Ablations on the components of the Open-Det. "Losses" means the combination of the MAL and Joint Loss.

| ODR | BVLA-M | VLD-M | ONG | Losses | Eps | $AP_r$ | $AP_c$ | $AP_f$ | AP |
|---|---|---|---|---|---|---|---|---|---|
| | | | | | 31 | 10.2 | 17.4 | 23.2 | 19.6 |
| ✓ | | | | | 31 | 13.9 | 19.8 | 27.6 | 23.1 |
| ✓ | ✓ | | | | 31 | 14.7 | 20.3 | 27.9 | 23.5 |
| ✓ | ✓ | ✓ | | | 31 | 16.3 | 24.2 | 29.9 | 26.3 |
| ✓ | ✓ | ✓ | ✓ | | 31 | 16.9 | 24.5 | 29.7 | 26.3 |
| ✓ | ✓ | ✓ | ✓ | ✓ | 31 | 21.0 | 24.8 | 30.1 | 27.0 |

**Ablations on the Losses.** We propose the *Masked Alignment Loss* to eliminate contradictory losses and the *Joint Loss* to enhance the binary classification. Table 4 indicates that both losses contribute to performance improvements, particularly for rare classes with limited training samples: the Masked Alignment Loss increases $AP_r$ by **+1.7%**, while the Joint Loss boosts $AP_r$ by **+2.6%**. When the two losses are combined, the model achieves the best performance, with significant gains of **+4.1%** in $AP_r$ and **+0.7%** in AP.

**Ablations on Object Name Generator Components.** We conduct ablation studies on the LoRa Head and Text Denoising training approach, both designed to enhance LLM convergence efficiency. As presented in Table 5, each component individually improves performance, especially for rare classes. When combined, they achieve optimal results

Table 4: Ablations on the proposed loss functions.

| Masked Alignment Loss | Joint Loss | $AP_r$ | $AP_c$ | $AP_f$ | AP |
|---|---|---|---|---|---|
| | | 16.9 | 24.5 | 29.7 | 26.3 |
| ✓ | | 18.6 | 24.2 | 30.1 | 26.6 |
| | ✓ | 19.5 | 23.9 | 30.0 | 26.5 |
| ✓ | ✓ | 21.0 | 24.8 | 30.1 | 27.0 |

than baseline, with $AP_r$ and AP increasing by **+5.7%** and **+1.0%**, respectively, confirming their effectiveness.

Table 5: Ablations on elements of *Object Name Generator*.

| LoRa Head | Text Denoising | $AP_r$ | $AP_c$ | $AP_f$ | AP |
|---|---|---|---|---|---|
| | | 15.3 | 23.5 | 30.1 | 26.0 |
| | ✓ | 16.5 | 24.5 | 29.9 | 26.4 |
| ✓ | | 19.9 | 23.2 | 29.9 | 26.2 |
| ✓ | ✓ | 21.0 | 24.8 | 30.1 | 27.0 |

### 4.3. Improvements in VL Alignment Scores

The Open-Det framework enhances Vision-Language (VL) alignment through two innovative components: the BVLA-M and the VLD-M. The BVLA-M focuses on improving query-text alignment, while the VLD-M distills image-text knowledge from a pre-trained VLM into aligned VL-prompts, enabling more accurate object name generation. To quantify the improvements in VL alignment, we compared the alignment scores for Open-Det and GenerateU on the LVIS MiniVal dataset (over 50,000 object instances) under a zero-shot transfer setting. As shown in Fig. 8, Open-Det achieves a higher alignment score of **0.555 ± 0.074**, significantly outperforming GenerateU (0.448 ± 0.026). This demonstrates the superior capability of Open-Det in achieving robust VL alignment.

Furthermore, Open-Det outperforms GenerateU (as shown in Table 1), which can be further supported by the higher similarity score between the generated texts and ground truth category names, as illustrated in Fig. 9 of the Appendix. Specifically, Open-Det achieves a similarity score of 0.628 ± 0.067, while GenerateU scores 0.620 ± 0.077. This demonstrates that the VLD-M distillation method for VL-prompts in Open-Det is highly effective, enabling the LLM to generate more precise and accurate object names.

### 4.4. Visualization and Analysis

As an efficient OED framework, Open-Det provides significant advantages by detecting potential objects for any images without requiring additional vocabulary priors, making it highly practical and applicable in real-world scenarios. In Fig. 4, we visualize the detection results on the LVIS MiniVal data. Compared to human-labeled annotations and

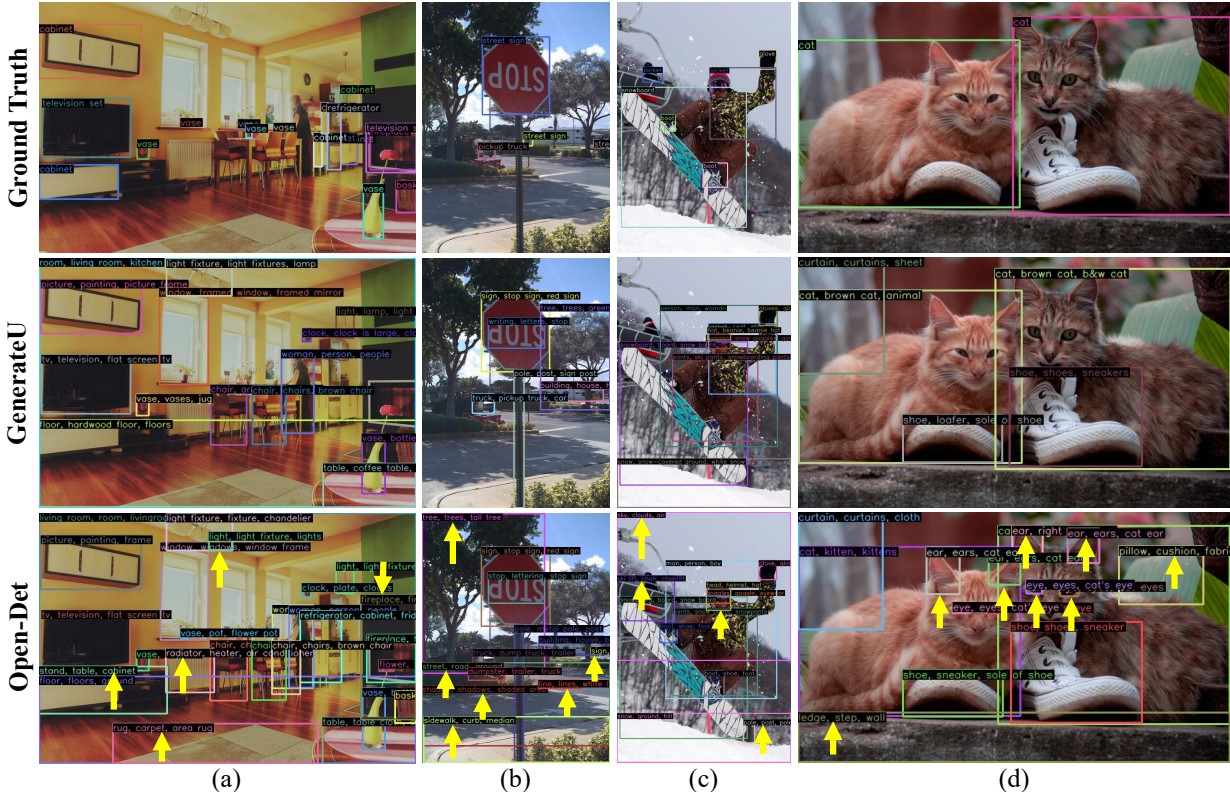

Figure 4: Visualization results for Ground Truth, GenerateU, and Open-Det on the LVIS MiniVal dataset. Open-Det demonstrates superior capability in detecting a broader range of potential objects in images (indicated by yellow arrows), covering large-scale objects, small objects, and fine-grained details, such as cabinet, rug, light, radiator, and fireplace in (a); sidewalk, shadows, street, and small sign in (b); and sky, sky lift, head, and pole in (c); wall, ear, eye, and pillow in (d).

the results of GenerateU, our model shows the superior capability to detect a broader spectrum of potential objects and accurately identify objects across varying sizes and granularity in the images, effectively highlighting its advanced capabilities and overall effectiveness. More visualizations and analysis are presented in Sec. C.2 of the Appendix.

### 4.5. Limitations and Future Works

Similarly to the existing OED framework of GenerateU, Open-Det's performance is primarily constrained by cross-modal semantic discrepancies between visual regions and image-like textual embeddings. These discrepancies arise from the interactions among the backbone, the detector, the VLM, and the LLM. To mitigate this limitation, employing stronger foundation models, such as InternImage (Wang et al., 2023) and Strip-MLP (Cao et al., 2023) for vision backbone models, ALIGN (Jia et al., 2021b) and CogVLM (Wang et al., 2024) for VLM models and DeepSeek (Guo et al., 2025) for generative language model, is an efficient method for further performance improving. Additionally, training on supplementary datasets can serve as an effective approach to enhance performance.

Furthermore, integrating a segmentation decoder module into Open-Det framework will offer mask priors for more precise regional and semantic feature extraction. This enhancement can further address semantic gap in cross-modal representations, transforming Open-Det into a unified detection-segmentation framework and boosting performance in both tasks.

### 5. Conclusion

This paper presents Open-Det, a novel and efficient generative framework for the OED task. It accelerates the model training for both the box detector and the LLM through the specifically designed architecture and enhancing region-text alignment between Vision and Language modalities. This is achieved through two key components: the Bidirectional Vision-Language Alignment module and the Vision-to-Language Distillation module, incorporated by the proposed Masked Alignment Loss and Joint Loss to further improve training efficiency and performance. In summary, Open-Det advances OED research by offering a robust and efficient framework, paving the way for future exploration of more flexible and practical OED approaches.

## Acknowledgements

This work was supported in part by the National Key Research and Development Program of China (Grant No. 2021YFF1200800), in part by the National Natural Science Foundation of China (Grant No. 62276121), in part by the National Natural Science Foundation of China (Grant No. 62402252), in part by the TianYuan funds for Mathematics of the National Science Foundation of China (Grant No. 12326604), in part by the Shenzhen International Research Cooperation Project (Grant No. GJHZ20220913142611021), and in part by the Pengcheng Laboratory Research Project (Grants No. PCL2023A08 and PCL2024Y02).

## Impact Statement

This paper introduces a novel and efficient Open-Ended Detection framework that significantly accelerates model convergence, enhances training efficiency and performance, and reduces reliance on large-scale datasets and high GPU resources for training. This enables practical and flexible vocabulary-free open-world detection. These advancements hold significant potential for real-world applications, such as autonomous driving and security, while also advancing research in Open-Ended Detection.

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

# A. Appendix for Method.

## A.1. Main Pipeline of the Open-Det Framework

In Fig. 2 of the main text, we present the main architecture of Open-Det. To further clarify its workflow, we provide a simplified pipeline in Fig. 5. During training, the input image is processed by the Detector to generate queries, which predict bounding boxes and binary classification (positive and negative samples) results. Simultaneously, the text corresponding to the objects in the image is fed into a frozen text encoder of the Vision-Language Model (VLM) to produce text embeddings. Then, Open-Det distills Vision-Language alignment knowledge from the VLM into queries, generating VL-prompts. These VL-prompts, along with text embeddings augmented with Gaussian noise, are input into the Large Language Model (LLM) for text reconstruction. Finally, the LLM automatically generates the names of the objects detected by the box Detector.

The Text Denoising training approach improves the LLM's training efficiency and robustness. The distillation module bridges the semantic gap between VL-prompts and text embeddings, reducing reliance on large-scale datasets through prompt distillation. The LoRa Head in the LLM reduces trainable parameters, further enhancing the overall training efficiency. As a result, our model achieves significantly higher performance than GenerateU while being trained on just 4 V100 GPUs (compared to 16 A100 GPUs used by GenerateU).

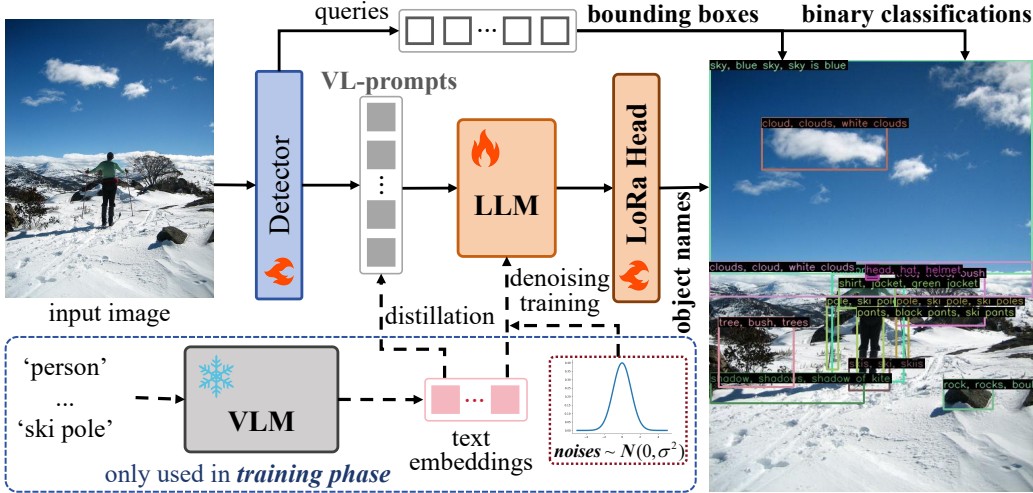

Figure 5: The simplified pipeline of the Open-Det framework. The Vision-Language Model (VLM) model and input texts are only used in the training phase. The symbols 🔥 and ❄️ represent that the model weights are activated and frozen, respectively.

## A.2. Effectiveness of the LoRa Head in the LLM

Generative language models, such as T5, often feature a heavy-weight head, making training challenging. For example, in the popular T5 (Raffel et al., 2020) model, the head layer responsible for generating object names contains **24.7M** parameters, mapping an input dimension of 768 to an output dimension of 32,128 (the vocabulary size of T5). By replacing the standard head with a LoRa Head, the number of trainable parameters is reduced to **0.526M**, which is only **2.12%** of the original parameter count. Additionally, the original pre-trained weights of the LLM head are frozen, preventing disruption when noisy training occurs in the early stages. This significant reduction in trainable parameters of LLM Head enhances computational efficiency while maintaining model performance.

## A.3. Causes of Contradictory Alignment Loss

In Sec. 3.6 of the main text, we analyze the contradictory losses that arise in the existing query-text alignment loss (Lin et al., 2024) for the OED task. In this section, we provide a detailed explanation of how these contradictory losses and their associated gradients are generated.

The existing alignment loss in GenerateU is calculated using the binary cross-entropy (BCE) loss, based on the similarity

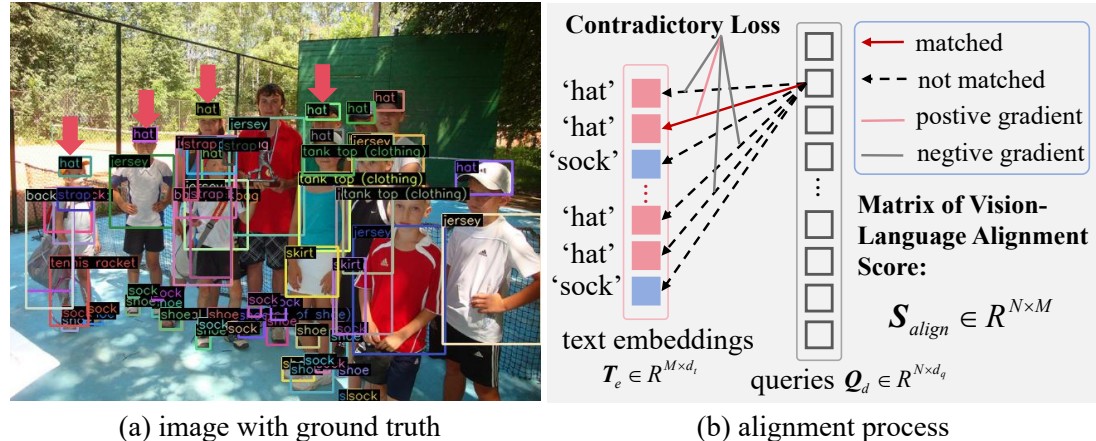

(a) image with ground truth        (b) alignment process

Figure 6: Illustration of contradictory loss generation in the query-text alignment process.

between the query feature and text embeddings from a fixed CLIP (Radford et al., 2021) text encoder. For each object query, *the corresponding word in the text encoder is treated as a positive sample, while all other words in the same mini-batch are treated as negative samples*. This arrangement establishes a contrastive learning framework, where the model maximizes the similarity between the query and its positive text embedding while simultaneously minimizing the similarity with negative text embeddings.

However, this formulation introduces a potential conflict in the optimization process, leading to **contradictory losses**. As shown in Fig. 6 (a), the image contains multiple instances of the same category, such as some people wearing *hats*. For the alignment loss in GenerateU, as illustrated in Fig. 6 (b), only one text embedding is matched with each single query (indicated by the red line as positive sample), while all other text embeddings are treated as unmatched (represented by black dashed lines as negative samples). Consequently, the query generates a ***positive*** loss with the matched text embedding for *hat*, while simultaneously generating ***negative*** losses with multiple unmatched text embeddings (including *hat*). However, the text embeddings generated by the CLIP encoder for the same text (like "*hat*") are identical. Therefore, the contradiction arises because the gradients computed for the query feature during back-propagation are influenced by two opposing forces:

- **Positive Gradient**: The gradient from the positive sample (the corresponding text embedding) encourages the query feature to move closer to the positive text embedding in the latent space.

- **Negative Gradients**: The gradients from the negative samples (all other text embeddings in the mini-batch) push the query feature away from the negative text embeddings.

In summary, when positive and negative text embeddings are semantically similar (*e.g.*, multiple instances of "*hat*"), contradictory alignment loss arises, resulting in opposing gradients. This conflict leads to unstable updates and suboptimal alignment. To address this issue, we introduce the Masked Alignment Loss, which corrects the labels to avoid generating contradictory losses. Experimental results in Table 3 and Table 4 demonstrate its effectiveness.

### A.4. Joint Loss: Effectiveness Analysis and Comparisons with Focal Loss and BCE Loss

As discussed in Sec. 3.6 of the main text, the positive and negative attributes of detected bounding boxes are predicted by a binary head. During training, the binary classification label for each query is *dynamically* determined through the matching process with ground truth boxes: matched queries are labeled as positive samples, while unmatched queries are labeled as negative samples. This dynamic matching process is determined by the cost of the GIoU (Rezatofighi et al., 2019), the L1 distance between predicted boxes and ground truth boxes, and the alignment score between queries and text embeddings. During model training, *the matching status of each query can dynamically shift between matched and unmatched*. This fluctuation may result in conflicting supervision, generating contradictory gradients and causing unstable training.

To address this issue, we propose a novel Joint Loss, defined in Eq. 4, which integrates the IoU score $u_i$, Alignment Score $s_i$, and the binary classification score $p_i$. To further illustrate its effectiveness, Fig. 7 visualizes the positive weights, negative

weights, and Joint Loss in relation to variations in the IoU score, Alignment score, and binary classification probability.

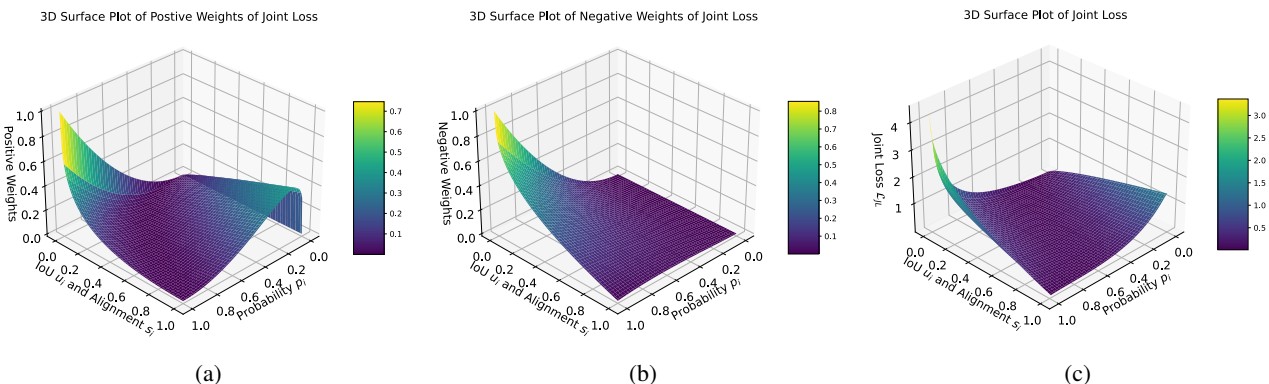

Figure 7: 3D surface plots of (a) positive weights, (b) negative weights, and (c) Joint Loss. To simplify the visualization in 3D space, we assume that the IoU score $u_i$ and the alignment score $s_i$ in Joint Loss have the same value (*i.e.* $u_i = s_i$) for queries.

In Fig. 7, we simplify the three-dimensional problem ($p_i$, $u_i$, $s_i$) to a two-dimensional plane by assuming $u_i = s_i$. This assumption implies that the model exhibits a consistent correlation between the predicted bounding boxes and the target categories, meaning that the localization accuracy (IoU) and semantic alignment (alignment score) are aligned (for convenience, we refer to these two scores as the IA Score). This simplification facilitates easier visualization and analysis of the relationships among the variables. Specifically, the effectiveness of our Joint Loss can be summarized as follows:

- **Positive Weights:** As shown in Fig. 7 (a), when the IA Score and probability $p_i$ are close (or ***consistent***) in value, it indicates that the binary head accurately predicts the binary classification, aligning well with both localization accuracy and semantic alignment. In this scenario, the positive weight remains relatively small (in the diagonal area at the center of the figure), which contributes to training stability. This consistency allows the model to quickly adjust its gradients and converge when the matching state changes. Conversely, when the IA Score or probability is significantly higher than the other—indicating ***inconsistent***—it suggests an incorrect probability prediction. This inconsistency stimulates an increase in positive weights (in both the left and right corners of the figure), thereby enhancing the loss of weight for positive samples.

- **Negative Weights:** During training, positive samples are typically *scarce*, whereas negative samples are more *abundant*, leading to a significant imbalance of samples. This imbalance challenges the model to effectively detect the rare positive samples. In addition, *large negative weights can drive the binary head's predicted probability closer to 0*, potentially further reducing the prediction of positive samples. To address this, we carefully design the negative weights in the Joint Loss, as illustrated in Fig. 7 (b): (1) When the probability $p_i$ is close to the IA score (indicating they are ***consistent***), it suggests that the binary head predicts an accurate binary classification score, regardless of whether the current dynamic matching is correct or not. Specifically: if both scores are high or low (along the diagonal in the figure), the probability prediction is relatively accurate; if the IA score is high but the probability score is low (in the right corner of the figure), it likely indicates an incorrect label, and the negative weights should be small to avoid introducing incorrect supervision. In both scenarios, small negative weights help mitigate training instability caused by label changes in dynamic matching and maintain the model's ability to detect positive samples. (2) Only when the IA score is low and the probability score is high (in the left corner of the figure), do the negative weights increase significantly. This indicates that the current matching and labels are ***correct***, but the binary head's prediction is ***incorrect***. In this case, the negative weights are increased to reduce the predicted probability score for the negative sample, improving the model's predictions.

- **Joint Loss:** The 3D surface plot of the Joint Loss is presented in Fig. 7 (c). From the plot, we can observe that: (1) When the IA score and the probability score are ***consistent*** (along the diagonal of the figure), it indicates accurate probability predictions, resulting in a smaller loss. (2) In contrast, when they are ***inconsistent*** (in both the left and right corners), it suggests inaccurate probability predictions, leading to a larger loss.

In summary, the Joint Loss effectively integrates the alignment score, IoU score, and probability score, mitigating incorrect supervision and training instability caused by dynamic matching. This results in a significant improvement in model performance, as demonstrated in Table 3 and Table 4 of the main text. Both theoretical analysis and experimental results validate the effectiveness of the Joint Loss.

**Differences from Focal Loss and BCE Loss:** Compared to Focal loss (Lin et al., 2017) and BCE loss, the proposed Joint loss adjusts the binary loss weights using consistent weights by associating the IoU score, alignment score, and binary score (), adaptively generating a "*soft loss*". For positive samples, the weight increases as the difference between the binary score and the *IA Score* grows; for negative samples, the weight increases only when the IA Score is notably high. This design effectively mitigates the negative impact of noisy labels from matched queries during dynamic matching and benefits rare-class detection, enhancing both training efficiency and accuracy. In contrast, Focal Loss and BCE Loss are limited to computing a "*hard loss*" based solely on the matching results of queries. This hard loss can lead to contradictory supervisory and gradients when the matching state of a query changes during the dynamic matching process in model training.

### A.5. Composition of Multiple Loss Functions

As presented in Fig. 2, the end-to-end OED framework of Open-Det consists of four collaborative components. The loss functions are also comprised of four parts: the loss from the *Object Detector*, the loss from the *Prompts Distiller*, the loss from the *Vision-Language Aligner*, and the loss from the *Object Name Generator*. The total loss can be formulated as:

$$\mathcal{L} = \mathcal{L}_{\text{ODR}} + \mathcal{L}_{\text{VLD}} + \mathcal{L}_{\text{VLA}} + \mathcal{L}_{\text{ONG}} \tag{5}$$

where $\mathcal{L}_{\text{ODR}}$ represents the loss associated with the *Object Detector* for bounding box generation; $\mathcal{L}_{\text{VLD}}$ denotes the distillation loss from the Vision-Language Model to the VL-prompts; $\mathcal{L}_{\text{VLA}}$ indicates the alignment loss between the query and the text embeddings; and $\mathcal{L}_{\text{ONG}}$ refers to the loss from the Large Language Model for generating object names.

The details of these loss functions are as follows:

(1) **The Object Detector Loss.** This loss is similar to those used in most transformer-based detectors (Carion et al., 2020; Zhang et al., 2022a; Hu et al., 2023; Lin et al., 2024), including the GIoU (Rezatofighi et al., 2019), L1 loss for bounding box generation. Additionally, it incorporates our proposed Joint Loss for positive and negative object predictions:

$$\mathcal{L}_{\text{ODR}} = \mathcal{L}_{\text{GIoU}} + \mathcal{L}_{\text{L1}} + \mathcal{L}_{\text{JL}} \tag{6}$$

(2) **The Vision-Language Distillation Loss.** This loss is the cosine similarity loss for mapping the query to its associate text-embedding:

$$\mathcal{L}_{\text{VLD}} = 1 - cosine(\boldsymbol{Q}_d, \boldsymbol{T}_e) \tag{7}$$

where $\boldsymbol{Q}_d$ and $\boldsymbol{T}_e$ represent the query features and text-embeddings, respectively.

(3) **The Vision-Language Alignment Loss.** This loss is the proposed Masked Alignment Loss:

$$\mathcal{L}_{\text{VLA}} = \mathcal{L}_{\text{MAL}} \tag{8}$$

(4) **The Object Name Generator Loss.** Similar to the language modeling loss used in GenerateU (Lin et al., 2024), we adopt cross-entropy loss to train its sequence-to-sequence architecture of the T5 (Raffel et al., 2020) model in our *Object Name Generator*. This loss measures the difference between the model's predicted token probabilities and the actual target tokens. It is a standard loss function for text generation tasks and is widely used in natural language processing (NLP). This loss can be formulated as:

$$\mathcal{L}_{\text{ONG}} = -\sum_{i=1}^{N} \sum_{j=1}^{V} y_{i,j} \log(\hat{y}_{i,j}) \tag{9}$$

where $N$ and $V$ represent the number of tokens in the output sequence and the size of the model's vocabulary, respectively. $y_{i,j}$ denotes the ground truth label for the $i$-th position in the output sequence and the $j$-th token in the vocabulary. It is a one-hot encoded vector where the correct token is 1, and all others are 0. $\hat{y}_{i,j}$ represents the predicted probability for the $i$-th position in the output sequence and the $j$-th token in the vocabulary. This is the probability assigned by the model to each token in the vocabulary.

# B. Appendix for Experiments.

## B.1. Details of Training Epochs

As shown in Table 1 of the main text, the results in rows 8, 9, and 11 correspond to the GenerateU and Open-Det models, both trained using *Iteration-based* training on the same VG (Krishna et al., 2017) dataset. However, due to differences in their training devices and batch sizes, directly comparing the number of iterations is not feasible. To enable a fair comparison, we convert iterations into epochs, providing a consistent measure of how many times each model has seen the entire dataset, independent of batch size or hardware differences.

Specifically, GenerateU (Lin et al., 2024) is trained using 16 A100 GPUs with a batch size of 64 for 180,000 iterations, while Open-Det is trained using 4 V100 GPUs with a batch size of 8 for 300,000 iterations. As presented in Table 6, Open-Det achieves higher performance with only **31** training epochs, which is **20.8%** of the epochs required by GenerateU (**149** training epochs). This highlights that our model significantly accelerates training convergence while maintaining superior performance.

Table 6: Training epochs for GenerateU and Open-Det.

| Model | Training Data | Image Size | Training Devices | Batch Size | Iteration | Epochs | $AP_r$ | $AP_c$ | $AP_f$ | AP |
|---|---|---|---|---|---|---|---|---|---|---|
| GenereteU (Lin et al., 2024) | VG (Krishna et al., 2017) | 77,398 | 16 A100 | 64 | 180,000 | 149 | 17.4 | 22.4 | 29.6 | 25.4 |
| **Open-Det (ours)** | VG (Krishna et al., 2017) | 77,398 | 4 V100 | 8 | 300,000 | **31** | **21.0**$_{\uparrow 3.6}$ | **24.8**$_{\uparrow 2.4}$ | **30.1**$_{\uparrow 0.5}$ | **27.0**$_{\uparrow 1.6}$ |

## B.2. Ablation Study: Freezing vs. Training LLM

The LLM is a core component of the Open-Det framework, enabling object category recognition and name generation. To assess its impact, we compare two training strategies for the OED task:

- Freezing the LLM weights (only activating the head);

- Activating all of the LLM weights.

As shown in Table 7, our Open-Det framework achieves superior performance compared to GenerateU in both training strategies, with improvements of **+5.6%** and **+3.6%** in $AP_r$, and **+7.7%** and **+1.6%** in AP, demonstrating consistent and significant superiority over GenerateU.

Notably, activating the LLM weights of Open-Det further boosts performance, yielding additional gains of **+10.5%** in $AP_r$ and **+7.0%** in AP. These performance stem from the domain shift between the image-text data used for LLM pre-training and the region-text alignment required for detection tasks. Experimental results demonstrate that Open-Det effectively enhances the region-text alignment, leading to superior overall performance.

Table 7: Ablation results on the training strategy of *Freezing* LLM vs. *Training* LLM. The symbol "†" indicates that the model was trained by us using the public official code of GenerateU, utilizing 4 V100 GPUs.

| Model | Backbone | LLM | Pre-train Data | Data Size | Epochs | $AP_r$ | $AP_c$ | $AP_f$ | AP |
|---|---|---|---|---|---|---|---|---|---|
| GenerateU† (Lin et al., 2024) | Swin-Tiny | Freeze | VG | 0.077M | 31 | 4.9 | 6.1 | 19.2 | 12.3 |
| **Open-Det(ours)** | Swin-Tiny | Freeze | VG | 0.077M | 31 | **10.5**$_{5.6}$ | **15.0**$_{8.9}$ | **26.1**$_{6.9}$ | **20.0**$_{7.7}$ |
| GenerateU (Lin et al., 2024) | Swin-Tiny | Train | VG | 0.077M | 149 | 17.4 | 22.4 | 29.6 | 25.4 |
| **Open-Det(ours)** | Swin-Tiny | Train | VG | 0.077M | 31 | **21.0**$_{\uparrow 3.6}$ | **24.8**$_{\uparrow 2.4}$ | **30.1**$_{\uparrow 0.5}$ | **27.0**$_{\uparrow 1.6}$ |

# C. Appendix for Results Visualization.

## C.1. Visualization of Query-Text Alignment Score and Text Similarity Score

For each matched query, we can compute the *query-text alignment score* and *text similarity score*, simultaneously. The query-text alignment score measures the cosine similarity between the query and the matched text embedding. The text similarity score evaluates the cosine similarity of text embeddings between the generated text and ground truth object names.

In Sec. 4.3 of the main text, we compare the Vision-Language alignment scores (query-text alignment) and text similarity scores of Open-Det and GenerateU (Lin et al., 2024). The results show that:

- **Vision-Language Alignment:** Open-Det achieves significantly higher query-text alignment scores than GenerateU, with scores of 0.555 compared to 0.448, as shown in Fig. 8. This represents a **+23.88%** improvement over GenerateU, confirming Open-Det's superior ability to align visual and textual information.

- **Text Generation Accuracy.** Open-Det also achieves higher text-embedding similarity scores between generated object names and ground truth object names, demonstrating its ability to generate more accurate object names.

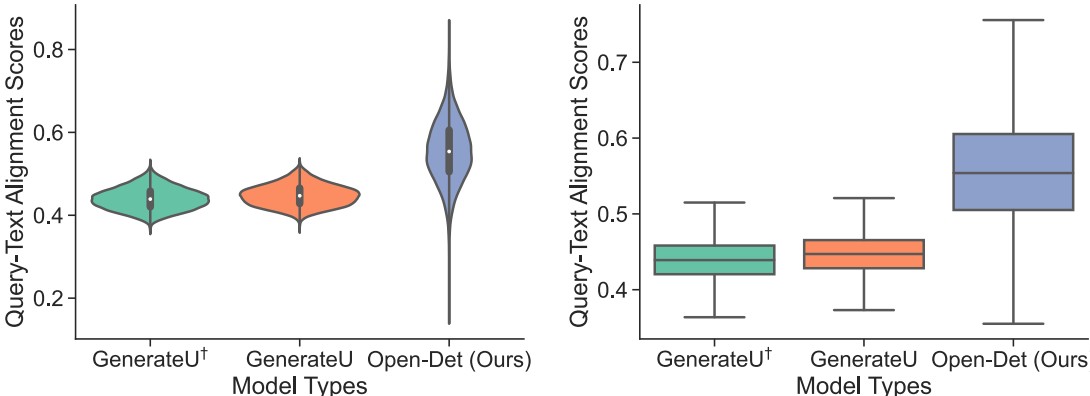

Figure 8: Comparison of VL alignment scores between GenerateU and Open-Det using violin and box plots.

In Fig. 9 (a), we visualize the alignment scores and text similarity scores for matched queries in a 2D space. The figure illustrates that Open-Det achieves higher alignment scores and text similarity scores, with points being more densely distributed. This suggests that Open-Det is more robust in both query-text alignment and object name generation.

In Fig. 9 (b), we further visualize these two scores alongside the Integrated Alignment-Similarity Score (the product of the two scores for each query) in a 3D space. The figure shows that most points for Open-Det are consistently positioned above those of GenerateU, further confirming our model's higher consistency and superior ability in accurate alignment and object name generation.

### C.2. Visualization and Analysis for Open-Ended Detection Results

Due to space constraints, Fig. 4 in the main text includes partial images of zero-shot domain transfer results on the LVIS MiniVal dataset. To further demonstrate the effectiveness of the Open-Det model, we provide additional visual comparisons in Fig. 10 and Fig. 11, showcasing results from *Ground Truth*, *GenerateU*, and *Open-Det*. These visualizations offer a comprehensive evaluation of the model's performance across various conditions, as detailed below:

- **Variety of Object Types:** The evaluation includes both *single* and *multiple* objects, covering specific categories such as *humans, animals, vehicles, buildings, food*, and various other objects.

- **Diverse Scenarios:** The results span a wide range of environments, including indoor and outdoor settings, as well as scenes featuring *skies, streets, oceans, static scenes, and dynamic scenes*.

- **Scale and Granularity:** The visualized objects vary in scale, from *small* to *large*, and include both *coarse-grained* and *fine-grained* object types.

Specifically, detection results that surpass GenerateU are marked with yellow arrows. Open-Det demonstrates superior performance in the following key aspects of Open-Ended Detection:

- **Superior detection ability from coarse-grained to fine-grained objects detection:** Open-Det is capable of detecting both coarse-grained and fine-grained objects, demonstrating superior region-text alignment compared to GenerateU.

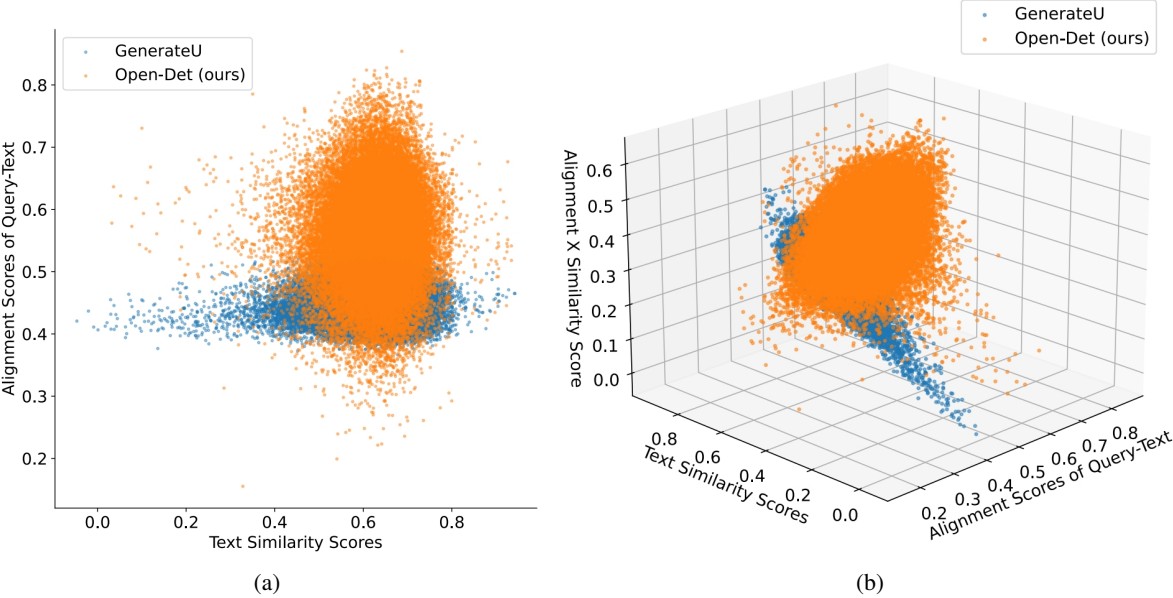

Figure 9: Visualization of query-text alignment score and text similarity score in zero-shot domain transfer on LVIS MiniVal dataset. (a) Visualization of alignment scores and text similarity scores for each matched query in a 2D space. (b) Visualization of both scores along with the Integrated Alignment-Similarity Score in a 3D space.

For instance, as shown in Fig. 10 (a) Open-Det detects coarse-grained objects like large-scale grass and fine-grained details such as the bear's eye and mouth, outperforming GenerateU. Similar detection results can be observed in Fig. 10 (b) ∼ (h) and Fig. 11, across objects of humans, animals, vehicles, buildings, and so on.

- **Capability to detect more potential objects:** For example, as shown in Fig. 11 (e), Open-Det identifies various objects on the table, such as a lanyard, a pen, and even the woman depicted within the book cover, which is not detected by Ground Truth or GenerateU. Similar detection results can be observed in Fig. 10 and Fig. 11.

- **More accurate object name generation:** For example, in Fig. 11 (b), Open-Det generates more precise object names, such as detecting the yellow traffic line on the street as a "*yellow* line", and identifying the yellow school bus as a "school bus" and "yellow bus", providing more accurate and descriptive categories.

- **Enhanced detection accuracy for occluded and blurred objects:** We also observe that Open-Det excels in detecting occluded and blurred objects. Examples include: 1) In Fig. 10: the occluded elephant in (b), the blurred plants and occluded person in (e), the occluded car in (f), and the blurred shoes in (h); 2) In Fig. 11: the occluded skis (with more accurate detection boxes than Ground Truth and GenerateU) in (f) and the occluded person in (h). These instances are highlighted with red arrows for clarity.

In conclusion, our Open-Det framework demonstrates several key advantages: faster training convergence, higher training efficiency with smaller datasets and fewer GPU resources, and superior overall performance. These strengths, supported by the accompanying visualizations, highlight the model's ability to automatically detect a wide range of potential objects from coarse-grained to fine-grained without requiring additional vocabulary priors. This capability holds significant practical value for real-world applications, such as autonomous driving, security, and intelligent transportation systems.

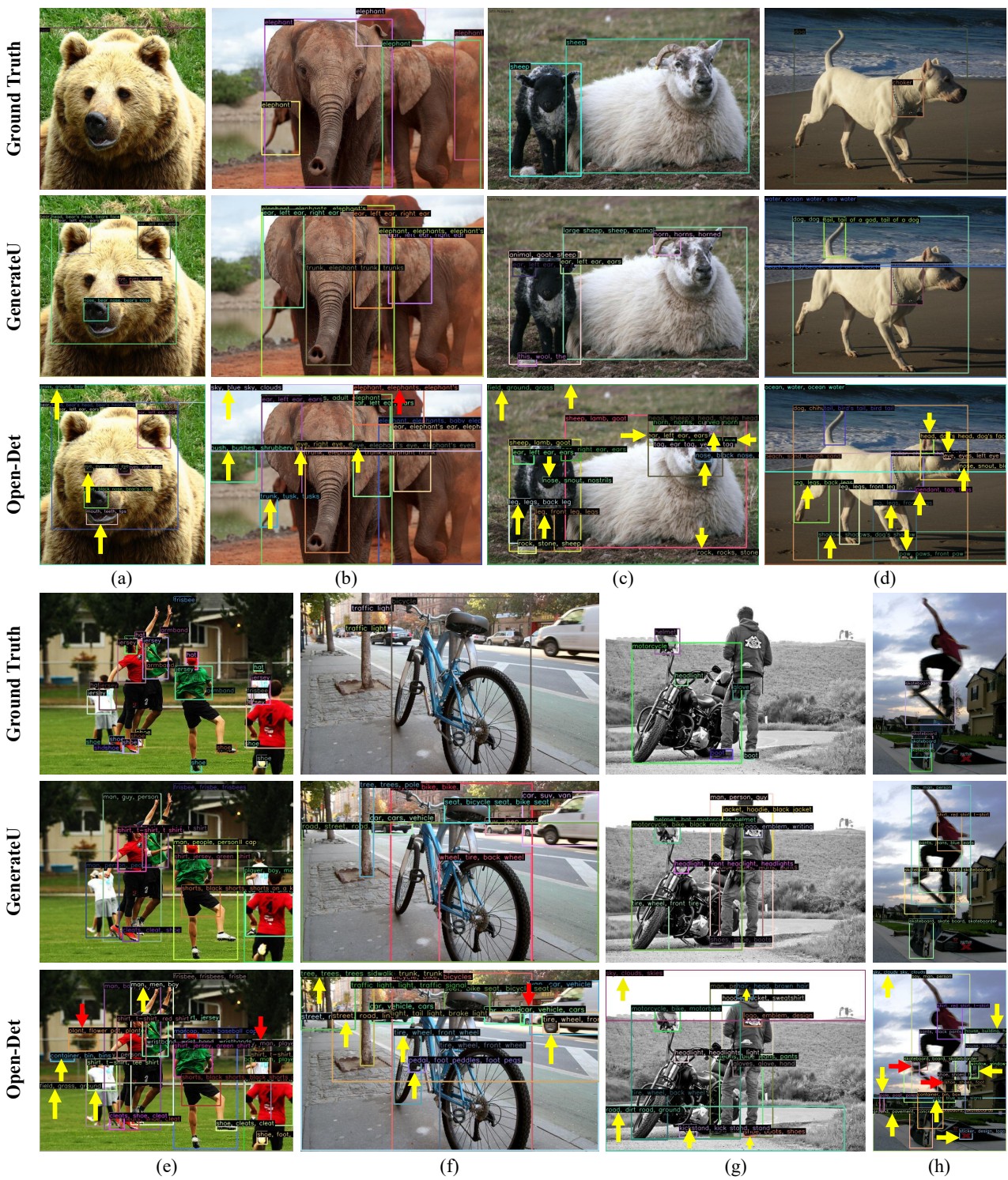

Figure 10: Visualization of detection results for Ground Truth, GenerateU, and Open-Det models on the LVIS MiniVal dataset. The scenes feature a variety of object types and levels of granularity.

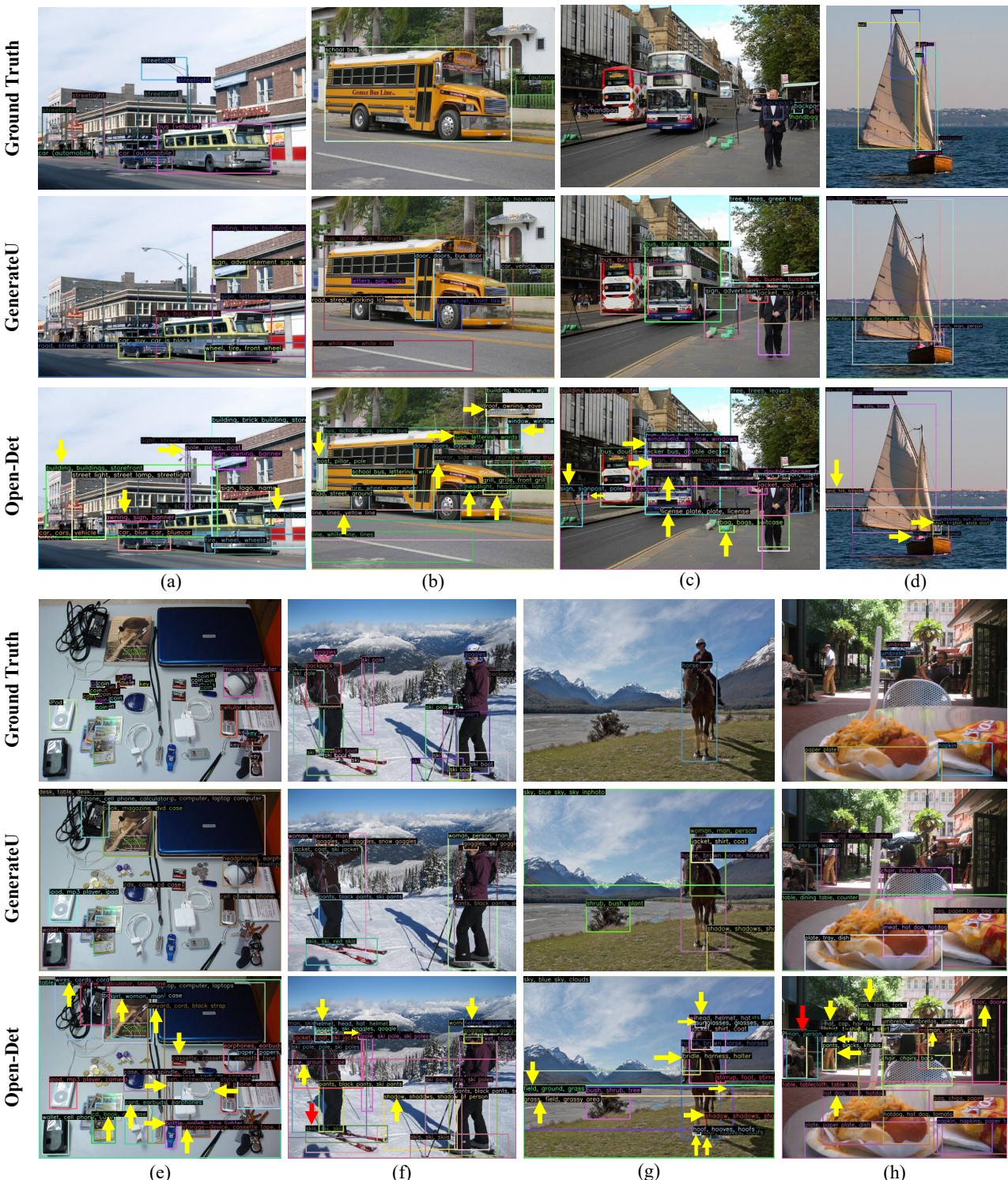

Figure 11: Visualization of detection results for Ground Truth, GenerateU, and Open-Det on the LVIS MiniVal dataset. The results feature diverse scenarios, such as streets, oceans, skies, and so on.

