# OpenReview forum: "Open-Det: An Efficient Learning Framework for Open-Ended Detection"
_ICML.cc/2025/Conference — ICML 2025 poster_

### Official Review · Reviewer_JNJc · 2025-03-06

**Overall Recommendation:** 4

**Summary:**

In this paper, the authors present a novel learning framework for open-ended detection. Open-ended detection consists in detecting objects that are not known a priori. The proposed framework is based on four main components: 1) an open object detector, 2) a prompt distiller, 3) an object name generator and 4) a vision language alignment module. the model is trained with a Visual Genome dataset (about 77K images). The pproposed model is compared to GenerateU, the SOTA model in open-ended detection, a new task in Deep Learning based Computer Vision. The proposed model slightly outperforms GenerateU in terms of AP. Moreover, Open-Det is more efficient in terms of training convergence (5 less epochs than GenerateU) and training data. The ablation study shows the effectiveness of each component of the proposed model. The paper is well written and the experiments are well conducted. This paper is a good contribution.

**Claims And Evidence:**

The authors claim that the proposed model is more efficient than GenerateU in terms of training convergence and size of training data. The authors provide evidence that the proposed model outperforms GenerateU in terms of AP. The ablation study shows the effectiveness of each contribution of the proposed model.

**Essential References Not Discussed:**

To my point of view, OED is close to Open Word Object Detection (OWD). Joseph, K. J., Khan, S., Khan, F. S., and Balasubramanian, V. N. Towards open world object detection. In Proceedings of the IEEE/CVF Conference on Computer Vision and Pattern Recognition (CVPR), pp. 5830–5840, June 2021.

It could be interesting to cite OWD in the related work section.

**Experimental Designs Or Analyses:**

/

**Methods And Evaluation Criteria:**

/

**Other Comments Or Suggestions:**

/

**Other Strengths And Weaknesses:**

++ Strong Points:
- The paper is well written and the experiments are well conducted.
- The proposed model is more efficient than GenerateU in terms of training convergence and size of training data.
- The ablation study shows the effectiveness of each component of the proposed model.
- There are many good insights in the paper: 1) a variable number of queries from encoder tokens in the object detector part, 2) the vision-language alignment module with the bidirectional alignment, 3) the joint loss function that  merges binary score with IoU and alignment score. 3) improves the performance for rare objects by 4%.


-- Weak Points:

- Although the proposed model is more efficient than GenerateU, the performance gain is not significant. The authors should provide more insights on the limitations of the proposed model and how to improve it.
- In the GenerateU paper, the authors achieved a better performance by using a Swin-L model. Why the authors did not use this model in order to show if Open-Det can also benefit from this BB and outperform GenerateU?
- The training data is small (77K images). Did you try to train the model with a larger dataset and to see if the performance is improved?

**Questions For Authors:**

Did you try to train the model with a larger dataset and to see if the performance is improved?

**Relation To Broader Scientific Literature:**

/

**Theoretical Claims:**

/

---

> ### Author Rebuttal · Authors · 2025-04-01
>
> Dear reviewer JNJc,
>
> We sincerely appreciate your valuable feedback and have incorporated all suggestions into the manuscript for the next version.
>
>
> **Q1: OED is close to OWD. It could ... section.**
>
> We have carefully checked the OWD [1], which enables the detector to label unknown objects as ''unknown'' and incrementally learn these identified unknown categories without forgetting previously learned classes. In contrast, our method could directly detect novel/unseen categories without requiring any prior human knowledge. In other words, both OWD and OED are capable of detecting and recognizing new categories, demonstrating their relevance. Following your suggestion, we will include this work in the related work section in the new version.
>
> **Q2: The authors ... insights ... improve it.**
>
> Similarly to the existing OED framework of GenerateU, Open-Det's performance is primarily constrained by cross-modal semantic discrepancies between **visual regions** and image-like **textual embeddings**. These discrepancies arise from the *interactions* among the detector, the VLM, and the LLM. To mitigate this limitation, employing stronger foundation models, such as ALIGN and CogVLM-17B for VLM models and DeepSeek for generative language model, is an efficient method for further performance improving. Additionally, training on supplementary datasets can serve as an effective approach to enhance performance.
>
> Furthermore, integrating a segmentation module into Open-Det framework will offer mask priors for more precise **regional** and semantic feature extraction. This enhancement can further address semantic gap in cross-modal representations, transforming Open-Det into a unified **detection-segmentation** framework and boosting performance in both tasks. We will incorporate this discussion in the new version and propose it as a direction for future research.
>
>
>
> **Q3: Why the ... Swin-L model ... GenerateU?**
>
> Compared to Swin-T, the Swin-L model has a significantly larger number of parameters (**197M vs. 29M, a difference of 6.8$\times$**) and supports a higher input image size (384 $\times$ 384 vs. 224 $\times$ 224, with GenerateU using a pretrained Swin-L at a resolution of 384 $\times$ 384). This results in a substantial increase in GPU memory usage during model training (e.g., Swin-L requires approximately 4 $\sim$ 5 times more memory than Swin-T), making it **infeasible** for Open-Det to train on 4 V100 GPUs.
>
> It is fair and effective to evaluate the performance of GenerateU and Open-Det models with the **same backbone** (such as Swin-T). Therefore, we did not use Swin-L for performance comparisons in the first instance.
>
> However, as suggested, we managed to conduct **additional experiments** with two backbones: Swin-S (using 4 V100 GPUs) and Swin-L (using 4 A800 GPUs). The main experimental results are as follows:
>
> | Model | Backbone | Train Data | Data Size | Epochs | AP$_{r}$ | AP$_{c}$ |  AP$_f$ | AP |
> | :---: | :---: | :---: | :---: | :---: | :---: | :---: | :---: | :---: |
> | GenerateU | Swin-L | VG,GRIT5M | 5.077M | - | 22.3 | 25.2 | 31.4 | 27.9 |
> | **Open-Det (ours)** | Swin-T | VG | 0.077M | 31 | 21.0 | 24.8 | 29.8 | 27.0 |
> | **Open-Det (ours)** | Swin-S | VG | 0.077M | 31 | 26.0 (+3.7%) | 28.6 (+3.4%) | 32.8 (+1.4%) | 30.4 (+2.5%)|
> | **Open-Det (ours)** | Swin-L | VG | **0.077M** | 31 | **31.2 （+8.9%）** | **32.1 （+6.9%）** | **34.3 （+2.9%）** | **33.1 （+5.2%）** |
>
>
> **Results:**
> * Open-Det-Swin-S achieves an improvement of **+5.0%** in AP$_r$ (26.0% vs. 21.0%) and **+3.4%** in AP (30.4% vs. 27.0%) than Open-Det-Swin-T, demonstrating its efficiency.
>
> * Using only **1.5%**  training data, **Open-Det-Swin-S** outperforms *GenerateU-Swin-L* by **+3.7%** in AP$_r$ and **+2.5%** in AP.
>
> * When utilizing the larger backbone of Swin-L, **Open-Det-Swin-L** significantly outperforms *GenerateU-Swin-L* by ``+8.9%`` in AP$_r$（31.2% vs. 22.3%）and ``+5.2%`` in AP（33.1% vs. 27.9%）, further confirming its superior effectiveness and efficiency.
>
> We will include these results in the new version.
>
>
>
> **Q4: Did you try to ... is improved?**
>
> Intuitively, more training data would further improve the performance of Open-Det. Unfortunately, due to current limitations in available time (within the rebuttal days) and GPU resources, conducting experiments with the large-scale dataset (such as GRIT5M) presents significant challenges.
>
> As an implementable scheme, we further extended Open-Det's training on the VG dataset from 31 to 50 epochs to investigate potential performance improvements under practical resource limitations. The results show that extended training further improves the performance with **+0.9%** (21.9% vs. 21.0%) in AP$_r$ and **+0.4%** (27.4% vs. 27.0%) in AP, confirming its effectiveness. However, we believe it remains valuable to investigate the impact of training data scale on performance for furture studies when more GPU resources become available.
>
>
> [1] Towards open world object detection. CVPR 2021.

---

### Official Review · Reviewer_AMke · 2025-03-13

**Overall Recommendation:** 1

**Summary:**

This paper introduces **Open-Det**, which addresses the inefficiencies of previous open-ended methods, including slow training and high memory consumption. Open-Det achieves improved efficiency and performance through:

1. Enhancing the object detector,
2. A vision-language (VL) alignment module,
3. A VL distillation module,
4. LoRA for parameter-efficient fine-tuning,
5. A noise-adding and denoising strategy,
6. Masked Alignment Loss.

## update after rebuttal
During the rebuttal, the authors repeatedly used terms like “factual error” in an attempt to assert the originality of their method, which I find unconvincing. In my view, the proposed approach does not appear substantially different from prior work—though this may require further verification by other reviewers or the AC. That said, I believe there is nothing wrong with building upon existing methods; standing on the shoulders of giants is how we advance the community. However, in the method description section, the authors failed to acknowledge this connection. I trust that the proposed modules were likely inspired by previous work, and this omission is a notable weakness. I will maintain my original score and leave the final judgment to other reviewers and the AC.

**Claims And Evidence:**

The analysis and claim of **GeneratU** in the paper is clear and convincing.

**Essential References Not Discussed:**

Please see method section. I firmly believe that the vast majority of contributions presented in this paper were originally proposed and validated in the object detection domain. However, the authors neither discuss nor cite these prior works, which raises concerns about the completeness and transparency of their literature review.

**Experimental Designs Or Analyses:**

Missing results on COCO and Obje365, which is common Open-end dataset and evaluated in in GenerateU

**Methods And Evaluation Criteria:**

**My biggest concern is that the proposed acceleration methods for open-ended detection are adaptations of techniques already introduced in traditional object detection, yet the authors fail to cite any of them. This omission gives the impression of misleading novelty.**

3.2 Object Detector: **a threshold-based query selection** is essentially the common two-stage approach found in [1][2][3] and so on.

3.2 Object Detector: *divide the first layer and last two layer* is similar to the idea presented in [4].

3.4. Vision-to-Language Distillation: *using deformable cross-attention to attend encoder feature* is exactly the same as [1]

3.4. Vision-to-Language Distillation: *using text-embedding to supervise query feature* is a common trick used in open-vocabulary (Almost every method uses this approach.)

3.5. Object Name Generator: *Noisy Alignment.* The noise-adding and denoising strategy is the same as in [2].

3.6. Masked Alignment Loss: *Mask text-classifier based on feature similarity* is the same as [5].

*The key issue is that the authors did not cite any of the aforementioned works, which raises serious concerns. Given that many of these methods are well-known, it is unlikely that the authors are unaware of them. Instead, it appears that they may have deliberately omitted these citations, which could be misleading.*

[1] Deformable Transformers for End-to-End Object Detection

[2] DINO: DETR with Improved DeNoising Anchor Boxes for End-to-End Object Detection

[3] Dynamic Anchor Boxes are Better Queries for DETR

[4] DAC-DETR: Divide the Attention Layers and Conquer

[5] Scaledet: A scalable multi-dataset object detector

**Other Comments Or Suggestions:**

N/A

**Other Strengths And Weaknesses:**

N/A

**Questions For Authors:**

N/A

**Relation To Broader Scientific Literature:**

N/A

**Theoretical Claims:**

N/A

---

> ### Author Rebuttal · Authors · 2025-04-01
>
> Dear reviewer AMke,
>
> We thank your time and feedback. However, we respectfully disagree with comments (Q1 to Q7) and the assertions（the omitted citations and misleading novelty) for the following reasons:
>
> **Q1: The ..., yet ... cite ... novelty.**
>
> (1) **Reviewer's Factual Error:** The reviewer claimed ***3 times***: we **failed to cite any** of the 5 specified papers. In fact, we have explicitly **cited** ``3`` out of 5 in **multiple places** in main paper: Deformable DETR [1] (*Line 338*), DINO [2] (*Lines 87, 132, 152, 295, 796*), and DAC-DETR [4] (*Lines 79, 155, 161, 295, 797*). The assertion that *we deliberately omitted citations - and thus misleading novelty* - has ``no ground and constitutes a factual inaccuracy requiring correction``.
>
> (2) **Exclusion of [3,5]:** DAB-DETR [3] differs from Open-Det in *task* and *method* (e.g., Q2). Similarly, ScaleDet [5] tackles label unification for multiple-dataset training, differing from our focus in **Q7**. We initially excluded them but may be included in Related Works if space permits.
>
> (3) Open-Det enhances detection across object granularities (coarse-to-fine) and scales (large-to-small) (not solely in training acceleration). Notably, it demonstrates *additional vocabulary priors are not necessary in inference*, with ***11.7%*** training data achieving **+6.8%** AP$_r$ and **+8.5%** AP over GLIP(A) of OVD. We believe our work establishes a **foundational framework** for future OED research.
>
> **Q2: Object is ... found in [1][2][3].**
>
> **Reviewer's Factual Error:**
> * **[1]**: Obtains Content Query (CQ)/Positional Query (PQ) via Top-K selected sinusoidal-encoded anchors (**no threshold**);
>
> * **[2]**: Initializes CQ statically, derives PQ from Top-K boxes (**no threshold**);
> * **[3]**: Uses static initialization for both CQ/PQ (**no threshold**)
>
> **Innovation of Open-Det:** Different from [1-3], Open-Det introduces a NOVEL single adaptive query type (rather than CQ and PQ), transforming fixed number of objects prediction into flexible.
>
>
> **Q3: Object ... DAC-DETR.**
>
>
> * **Reviewer's Factual Error:** DAC-DETR [4] employs a dual-decoder architecture (standard O-Decoder and auxiliary C-Decoder) to enhance training efficacy, rather than dividing the first and last two decoder layers.
>
> * **Advantages of Open-Det:** In contrast, our Open-Det directly optimizes the architecture of standard single-decoder (Sec.``3.2``), enabling lower training costs and superior efficiency.
>
>
> **Q4: Vision ... deformable attention (DA).**
>
> This claim **misinterprets** our contribution. We did **NOT** claim DA is our contribution. It is not suprise that DA is widely used. We also follow this practice. However, one of our novelty is the design of **VLD-M** (Fig.``3``), leveraging it to construct **image-like** queries for visual-text alignment. This represents both a **novel architectural design** and **distinct application**.
>
>
> **Q5: Vision ... supervise ...**
>
> * This claim without gives any references. **Text-embedding to supervise query feature is NOT a common trick but a priciple paradiam for vision-text alignment.** Exisiting mehtods differ in how the supervison is done, *rather than whether or not simply using it*. The reviewer's claim is NOT professional.
>
> * **Innovation of Proposed VLD-M:** Our VLD-M aims to adaptively construct **image-like** queries to reduce **region-text annotations** (Sec.``3.4``), improving AP$_c$ by **+3.9%** and AP by **+2.8%** (Table ``2``).
>
> **Q6: ONG: Noisy ... [2].**
>
> DINO adds *biased coordinates noise into box* to accelerate *box convergence*. However, Open-Det adds *Gaussian noise into text embeddings* of VLM to speed up *LLM*'s convergence. They **differ fundamentally** in both objectives (**box vs. LLM**) and noise types (**biased coordinates vs. Gaussian**).
>
> **Q7: MAL: ... ScaleDet [5].**
>
> * **Reviewer's Factual Error:** MAL is NOT used for text-classifier but for query-text alignment.
>
> * **Text Label Unification ≠ Multimodal Alignment:** They are **fundamentally different** in motivation and method. ScaleDet unifies text labels via semantic similarity (**as soft label in MSE**) to combine multi-dataset training, while MAL resolves query-text matching conflicts through **similarity-binarized BCE** updates (Sec.``3.6``).
>
> **Q8: Missing ... .**
>
> * We conducted test on COCO. Open-Det still outperforms GenerateU by **+2.8%** AP (35.8% vs. 33.0%).
>
> * Furthermore, additional experiment shows that Open-Det-SwinL surpasses GenerateU-SwinL by ``+8.9%`` in AP$_r$ on LVIS (see **Q3** responses to **JNJc**), demonstrating significant superiorities.
> ***
>
> According to our responses to Q1~Q8, our method is **NOT** an adaptation of existing techniques, but rather respresents a fundamentally different approach, innovating in framework, modules, loss functions, performance and efficiency.
>
>
> ``Overall, we respectfully request the reviewer to cross-check the citations, methodologies, as well as all our responses, and we appreciate a fair judgement.``

---

> > ### Comment · Reviewer_AMke · 2025-04-02
> >
> > The authors' feedback has reinforced my initial judgment. According to their response to Q1 ("The ..., yet ... cite ... novelty."), the authors clearly acknowledged being aware of the five papers I mentioned. However, they failed to appropriately cite three of these papers in relevant contexts (which constitutes misleading referencing) and intentionally omitted the other two. Regarding their replies to Q2–Q7, I perceive the authors as attempting to exaggerate minor differences. For instance, in Q2, does using a threshold selection method truly differ fundamentally from top-k selection? Similarly, in Q6, is replacing uniform noise sampling in DINO with Gaussian noise sampling genuinely a substantial distinction? Nevertheless, after carefully reviewing other reviewers' opinions, I have decided to maintain my current score and attribute my primary reason for rejection entirely to the paper's misleading claims regarding novelty and referencing. In my view, the paper still contains significant issues and does not meet the acceptance bar.

---

> > > ### Author Response · Authors · 2025-04-02
> > >
> > > We believe our paper has **NOT** been read carefully and has been subjected to unfounded speculative accusations regarding our intentions.
> > >
> > > Specifically, we have explicitly cited **3 of 5 papers** listed by reviewer AMke and referenced those papers in **mutiple places** in our ``initial manuscript`` (e.g., Deformable DETR [1] at *Line 338*, DINO [2] at *Lines 87, 132, 152, 295, 796*, and DAC-DETR [4] at *Lines 79, 155, 161, 295, 797*, totaling **11** ciations). If the reviewer AMke believes he is really familiar with our work, *it is unclear why he failed to notice that we, in fact, cited three of the five referenced papers with a total of 11 times* (Given initial comments ''*fail to cite any of them*'', which now has been reinterpreted by the reviewer AMke as "*failed to appropriately cite three of these papers*". This new comment is **contradictory to his initial comments** (which were repeated for 3 times) and **lacks any specific supporting facts**, which are **unconvincing** to us.). Are many essential contributions of our work also missed by reviewer AMke?
> > >
> > >
> > > In new feedback, reviewer AMke did **NOT** directly acknowledge these factual errors but presented new comment that we *constitute misleading referencing and intentionally omit the other two*. It is irresponsible to conclude that *we intentionally hid these popular references* based on **these factual errors**.
> > >
> > >
> > > Additionally, We would like to highlight that ``some factual errors and subjective ill-judged remarks appear once again in reviewer AMke's new feedback``. For example:
> > >
> > >
> > > * The new comment "*In Q2, does using a threshold selection method truly differ fundamentally from top-k selection?*" — In our response to Q2, we did NOT claim that "*threshold selection method fundamentally differs from Top-K selection*". The reviewer AMke initially claimed that ''*threshold-based query selection is found in [1][2][3] (Q2)*''. ``We just pointed out reviewer AMke's factual error`` that threshold selection method is **NOT** used in Deformable DETR [1], DINO [2], and DAB-DETR [3]. Notably, one of the novelties of our method is the proposed new single query type (differing from existing positional query and content query) and achieved flexible object prediction (differing from existing fixed query) for detection, but NOT the ''threshold'' as the key point. This novelty is also highlighted by the reviewer JNJc in Strong Points：good insights 1)... .
> > >
> > >
> > > * The new comment "*In Q6, is replacing uniform noise sampling in DINO with Gaussian noise sampling genuinely a substantial distinction?*" — In our response to Q6, we did NOT claim that "*the uniform noise sampling is substantial distinction to Gaussian noise sampling*" (it is reviewer's subjective ill-judegment). The denoising training is a common and effective training approach in deep learning. **The key is NOT which kind of *uniform noise sampling* or the *Gaussian noise sampling* were used**, but rather the *different problems being addressed* (closed-set detection vs. OED), *motivations* (box convergence vs. LLM's convergence), *method implementations in noise types* (biased coordinates (x, y, w, h) vs. text embedding vector with Gaussian noise). These core differences are overlooked by reviewer AMke once again.
> > >
> > >
> > > We believe we have adequately addressed the reviewer AMke's questions (with factual support, detailed analysis, and experimental results, etc., in our responses). Our Open-Det framework is novel in its *modules design (e.g., detector, BVLA-M, VLD-M, ONG), loss functions (MAL and Joint Loss), significantly improved performance (e.g., Open-Det-SwinL achieves an improvement of* **+8.9%** AP$_r$ and **+5.2%** AP *compared to GenerateU-SwinL), high efficiency* (using only* **1.5%** training data and **20.8%** training epochs than GenerateU), and resource superiority (**4 V100 GPUs** vs. **16 A100 GPUs** for training).
> > >
> > >
> > > Overall, ``we believe our paper has NOT been read carefully with convincing reviews`` and questions raised by reviewer AMke contain many factual errors in our opinion. At the same time, we do NOT think that the reviewer AMke's concerns (e.g., erroneous comments on citations) could weaken the core contributions of our paper. Given the current discussion, we find it difficult to reach a consensus. We will leave it for others to comment on publicly.
> > >
> > > Thank you for your efforts in reviewing the paper.

---

### Official Review · Reviewer_E2VT · 2025-03-13

**Overall Recommendation:** 4

**Summary:**

This paper presents Open-Det, a novel and efficient framework for Open-Ended Object Detection (OED), addressing the issues of slow convergence, low efficiency, and reliance on large-scale datasets found in existing models like GenerateU. Open-Det consists of four core components: the Object Detector (ODR), the Prompts Distiller, the Object Name Generator, and the Vision-Language Aligner. By introducing the Bidirectional Vision-Language Alignment module (BVLA-M) and the Vision-to-Language Distillation Module (VLD-M), Open-Det effectively bridges the semantic gap between vision and language. Additionally, the Masked Alignment Loss and Joint Loss further optimize training efficiency and classification consistency. Compared to existing models, Open-Det achieves superior performance using only 1.5% of the training data and 20.8% of the training epochs, while also detecting a broader range of objects, including smaller ones. Open-Det provides a highly efficient and scalable solution for OED tasks, with significantly reduced resource requirements.

**Claims And Evidence:**

- The claims made in the submission are generally supported by clear and convincing evidence. The paper provides experimental results that demonstrate the superiority of the Open-Det framework over existing models such as GenerateU and GLIP in various aspects, including performance, training data requirements, and convergence speed. Specifically, the evidence shows that Open-Det achieves higher performance with significantly fewer resources, requiring only 1.5% of the training data, 20.8% of the training epochs, and fewer GPUs, which is supported by quantitative results in the tables and performance curves.
- I have find a work named VL-SAM, they achieved 23.4 APr in LVIS, they are training free, so the resource and performance is better than yours.

**Essential References Not Discussed:**

Essential references are discussed.

**Experimental Designs Or Analyses:**

In the comparison experiments, the authors chose GLIP, which performs relatively poorly in OV detection, as a baseline. This seems somewhat unfair, as there have been many OV detectors that have achieved higher accuracy on LVIS after GLIP, such as the computationally intensive GroundingDINO or the more efficient CoDET, among others.

**Methods And Evaluation Criteria:**

- In the Open-Ended Detection task, objects can belong to multiple categories or have ambiguous semantics (e.g., "dog" vs. "pet"). How does Open-Det handle ambiguous or overlapping categories during inference, especially when there are multiple plausible names for a detected object? What is the impact of this ambiguity on the accuracy of object name generation, and how does the model ensure consistent results under these circumstances?
- The proposed Masked Alignment Loss (MAL) aims to address contradictory supervision, but how does this loss behave in practice when applied to images with multiple objects of the same category (e.g., several "cars" in a street scene)? Does the MAL genuinely prevent conflicting gradients, or could there still be scenarios where the supervision conflicts due to the inherent ambiguity in object detection tasks (e.g., detecting small objects in cluttered scenes)?
- The use of average precision (AP) and related metrics is standard, but how well do these metrics capture the true performance of Open-Det in open-ended object detection, where the focus is not just on detection but also on generating the correct category names? For example, how would Open-Det fare in tasks that require fine-grained distinctions between objects or the generation of complex object descriptions (e.g., detecting "a small red ball on the table" vs. "a ball")?

**Other Comments Or Suggestions:**

None.

**Other Strengths And Weaknesses:**

While Open-Det demonstrates impressive results with significantly reduced data, could the approach still face limitations in very low-data regimes, such as when there is an extremely low number of labeled examples for certain object categories? Does the framework struggle to generalize in scenarios where only a few instances of a new category are available, or when the object categories are underrepresented in the training set?

**Questions For Authors:**

None.

**Relation To Broader Scientific Literature:**

- The generative nature of Open-Det (which generates object names in addition to bounding boxes) relates to the work done on generative approaches in vision-language tasks, such as **image captioning** and **visual question answering (VQA)**, where models generate descriptive text based on visual inputs. The paper’s approach of using a **Large Language Model (LLM)** for name generation is in line with recent advancements in **transformer-based generative models** for language, like **T5** (Raffel et al., 2020).
- The paper incorporates a **Vision-to-Language Distillation Module (VLD-M)** for transferring knowledge from pre-trained Vision-Language Models (VLMs) into VL-prompts. where knowledge from a teacher model is transferred to a smaller or more efficient student model. In this case, the VLD-M distills alignment knowledge from powerful pre-trained VLMs (like CLIP) to improve the training of the Object Name Generator.

**Theoretical Claims:**

This work does not involve too much new mathematical theory or proofs.

---

> ### Author Rebuttal · Authors · 2025-04-01
>
> Dear reviewer E2VT,
>
> We deeply appreciate your suggestions and will include them into our revisions.
>
> **Q1: Training free VL-SAM .. better.**
>
> A direct performance comparison requires careful consideration of **model scales**:
>
> * **VL-SAM**: VLM: 17B, LLM: 7B, SAM: 0.632B, totaling ``24.632B``
> * **Open-Det**: VLM: 0.307B, LLM: 0.250B, Detector: 0.049B,  totaling ``0.606B``  (ONLY **2.46%**)
>
> With significantly **less** parameters (**40.6$\times$** ↓), Open-Det is still on par with VL-SAM in AP$_c$ (24.8% vs. 25.3%) and even **higher** in AP$_f$ (30.1% vs. 30.0%), indicating better trade-off between resouce and performance.
>
>
> **Q2: (1) How ... ambiguous? (2) What's ... accuracy? (3) How ... consistent results?**
>
> Language (text) inherently exhibits ambiguity, a characteristic **independent of the design of OED models**.
>
> (1) Similar to GenerateU, Open-Det also adopts ''*beam search*'' strategy to generate ``multi-level object names`` , effectively addressing the ambiguity issue and enhancing accuracy.
>
> (2) For fair evaluation, we adopt GenerateU's **standardized protocol**: comparing latent space similarities score between generated and annotated texts, *eliminating evaluation biases* from textual variability.
>
> (3) The adopted *beam search* strategy ensures consistent results, and Open-Det achieves higher text similarity scores (Appendix Sec. ``C.1``) than GenerateU.
>
>
> **Q3: (1) How  ... same category? (2) Does ... conflicting gradients, or (3) could  ... detection tasks?**
>
> (1) MAL transfers one-to-one matching (query to category, Fig. ``6``) to one-to-many matching (single query to multiple same-category instances) via:
>
> * **Computes text similarities** between all object-name pairs, offering a quantitative basis for dynamic label matching correction.
>
> * **Corrects dynamic label matching** (from 0 to 1) by transforming unmatched same-category pairs into matched ones (one-to-many) based on text similarities, resolving contradictory loss terms during optimization.
>
> (2) Yes, MAL efficiently addresses contradictory loss generation at its source.
>
> (3) MAL depends **solely** on textual ground truth, making it invariant to scene variations, but may be related to the annotations accuracy and text encoder.
>
>
> **Q4: (1) The use of AP ... category names? (2) how ...  descriptions (e.g., ...)?**
>
> (1) The evaluation assesses **both** box accuracy (IoU) and object naming performance (via cosine similarity (CS) score between **generated** and **annotated** texts). If object name is wrongly predicted, the CS score will be lower, thereby reducing the performance of AP.
>
> (2) Generated name granularity varies by training data annotations. Similar to GenerateU, Open-Det focuses on OED task, using **detection annotations** (i.e.,  object names and boxes) without utilizing *complex object descriptions*. Thus, Open-Det mainly predicts object categories (Fig.``4``, ``10``, ``11``), rather than fine-grained descriptions (i.e., obtaining relations between objects). We view the *task* of finer-grained name generation as an important area for future research.
>
>
> **Q5: The ... Grounding DINO or ... CoDET.**
>
> The comparison between Grounding DINO and Open-Det are as follows:
>
> | Model | Train Data | Data Size | AP$_{r}$ | AP$_{c}$ | AP$_f$ | AP |
> | :---: |  :---: | :---: | :---: | :---: | :---: | :---: |
> | Grounding-DINO-SwinT | O365,GoldG | 1.460M | 14.4 | 19.6 | 32.2 | 25.6 |
> | **Open-Det-SwinT(ours)** | VG | 0.077M | 21.0 | 24.8 | 30.1 | 27.0 |
>
>
> Using only **5.3%** training data, Open-Det-SwinT still significantly **outperforms** Grounding-DINO-SwinT by ``+6.6%`` in AP$_r$, ``+5.2%`` in AP$_c$,  and ``+1.4%`` in AP, further confirming its effectiveness and efficiency.
>
> CoDET [1] is designed for programming problems and has NOT been evaluated on object detection tasks. Due to fundamentally different objectives (**Code Generation** vs. **Object Detection**), a direct detection performance comparison is invalid. We also cross-check Co-DETR [2], which is designed for closed-set detection and can't be directly compared to our OED model.
>
>
> **Q6: (1) While ..., ... low-data regimes, such as ...? (2) Does ... training set?**
>
>
> (1) Like most OVD and OED models, Open-Det also faces common challenges related to low-data regimes and few-instance learning.
>
> (2) However, Open-Det demonstrates superiorities in both **efficiency** and **generalization** via VLM knowledge distillation into proposed VL-prompts. Notably, using significant **less** training data, it obtains superior performance on ``rare`` categories (namely only few instances of new category): **+6.8%** AP$_r$ over GLIP(A), **+6.6%** AP$_r$ over Grounding-DINO-SwinT, and **+3.6%** AP$_r$ over GenerateU. Using ``1.5%`` training data, Open-Det-SwinL outperforms GenerateU-SwinL by ``+8.9%`` in AP$_r$ (Please see our responses to **JNJc** in **Q3**)
>
>
> [1] Codet: Code generation with generated tests. ICLR 2023.
>
> [2] Detrs with collaborative hybrid assignments training. ICCV 2023.

---

### Official Review · Reviewer_EV8Z · 2025-03-19

**Overall Recommendation:** 3

**Summary:**

This paper proposes a new framework for open-ended detection. To address the problem of alignment between vision queries and text embeddings, the authors adopts a Bidirectional Vision-L;anguage Alignment module to obtain alignment score and uses a masked alignment loss to supervise the alignment beween queires and text embeddings. Additionally, a Vision-to-Language Distillation module is used to enhance the query feature to make more align to text embedding. The author conducts experiments on LVIS MiniVal dataset and show the proposed method can achieve better performances to GenerateU with the same pre-train data.

## update after rebuttal
The author's rebuttal address my concerns about the comparison with GenerateU. I keep my rating to Weak accept.

**Claims And Evidence:**

Yes.

**Essential References Not Discussed:**

No.

**Experimental Designs Or Analyses:**

Yes. I check the main experiments and ablation studies.

**Methods And Evaluation Criteria:**

Yes. Zero-shot domain transfer on LVIS MiniVal dataset is a popular choice for open set detection.

**Other Comments Or Suggestions:**

1. The Figure 2 is too complex and it not easy to understand the whole pipeline with Figure 2.

**Other Strengths And Weaknesses:**

Strengths

1. This paper proposes several reasonable modules to improve the performances of open-ended detection and conduct detailed ablation studies to validate the effectiveness of this modules. These modules contribute consistent performance improvement.
2. The idea of adopting IoU and alignment scores to enhance the loss function is interesting and the proposed approach inprove the performances significantly.
3. The proposed framework achieve better performances with less training cost comparing to recent SOTA methods.

Weaknesses
1. Can the proposed method achieve better performances with more training data? Comparing to GenerateU with VG and GRIT5M dataset, the proposed method trained with VG achieve not very significant performance improvement. It would be interesting to see the performance with VG and GRIT5M.

**Questions For Authors:**

Please refer to the Weaknesses in Other Strengths And Weaknesses.

**Relation To Broader Scientific Literature:**

The whole pipeline is similar to GenerateU which follows a pipeline of region proposal generator with a language model to generate object names for visual regions.

**Theoretical Claims:**

No. There are not theoretical claims.

---

> ### Author Rebuttal · Authors · 2025-04-01
>
> Dear reviewer EV8Z,
>
>
> We sincerely appreciate your time and constructive suggestions. We have carefully integrated your suggestions into the manuscript and will include these updates in the next version.
>
> **Q1: (1) Can the proposed ... more training data? (2) Comparing to ... improvement. It would be interesting to see the performance with VG and GRIT5M.**
>
> Thank you for your valuable suggestion.
>
> (1) Intuitively and empirically, it is commonly observed that larger datasets can enhance model accuracy in deep learning, as demonstrated in studies such as [1,2,3]. This principle is applicable to the Open-Det model as well.
>
> (2) There are two key reasons why Open-Det was not trained using the combined VG and GRIT5M datasets. **Firstly**, Open-Det's design prioritizes model efficiency, minimizing reliance on large datasets and training resources. Remarkably, when trained on the single VG dataset for only **20.8%** of GenerateU's training epochs, our model *outperforms* GenerateU which used both datasets. This result effectively demonstrates the effectiveness of our approach. **Secondly**, the GRIT5M dataset is **64.6$\times$** larger than VG dataset, which would require either:
>
> * **64.6$\times$** more GPU resources for the same training time (for instance, GenerateU was trained using *16 A100 GPUs*), or
> * **64.6$\times$** longer training time on our *4 V100 GPUs*
>
> These demands would result in substantial resource requirements and costs for model training. As a result, we excluded GRIT5M training from our initial submitted manuscript.
>
> Unfortunately, due to current limitations in available time (within the rebuttal days) and GPU resources, conducting experiments with the large-scale GRIT5M dataset presents significant challenges. However, following the reviewer's suggestion, we implemented a more computationally efficient alternative by extending Open-Det's training on the VG dataset from 31 to 50 epochs. This modified experimental design serves to:
>
> * Investigate potential performance improvements under practical resource limitations;
> * Validate the model's effectiveness without requiring additional combined dataset training;
>
>
> The experimental results are as follows:
>
> | Model | Backbone | Pre-Train Data | Data Size | Epochs | AP$_{r}$ | AP$_{c}$ | AP$_f$ | AP |
> | :---: | :---: | :---: | :---: | :---: | :---: | :---: | :---: | :---: |
> | GenerateU | Swin-T | VG | 0.077M | 149 | 17.4 | 22.4 | 29.6 | 25.4 |
> | GenerateU | Swin-T | VG,GRIT5M | 5.077M | - | 20.0 | 24.9 | 29.8 | 26.8 |
> | **Open-Det (ours)** | Swin-T | VG | 0.077M |  31 | 21.0 | 24.8 | 30.1 | 27.0 |
> | **Open-Det (ours)** | Swin-T | VG | 0.077M |  50 | **21.9** | **25.1** | **30.4** | **27.4** |
>
> Our extended training experiments yielded *consistent improvements* across all evaluation metrics. Compared to the 31-epoch baseline, additional training achieved: **+0.9%** improvement in AP$_r$ and **+0.4%** improvement in AP. Notably, Open-Det trained **solely** on VG dataset outperforms GenerateU (trained on both VG and GRIT5M), demonstrating **+1.9%** higher AP$_r$ and **+0.6%** higher AP. These comparative results provide further evidence of Open-Det's superior efficiency and effectiveness in the OED task.
>
> **Experiments on Larger Backbones:** In addition, when using larger backbones, such as *Swin-S* and *Swin-L*, Open-Det achieves significant improvements across all metrics. For instance, **Open-Det-Swin-S** obtains an improvement of **+5.0%** in AP$_r$ and **+3.4%** in AP than Open-Det-Swin-T. Using only **1.5%** training data, **Open-Det-Swin-L** *significantly outperforms GenerateU-Swin-L* by ``+8.9%`` in AP$_r$（31.2% vs. 22.3%）and ``+5.2%`` in AP（33.1% vs. 27.9%）, further confirming Open-Det's significantly superior effectiveness and efficiency. For more information, please refer to our responses to the Reviewer **JNJc** in **Q3**.
>
>
>
> **Q2: Figure 2 is complex and it is not easy to understand the entire pipeline.**
>
> Thank you for your valuable suggestion. Figure ``2`` may appear complex due to the *extensive detailed information* it contains. In Appendix ``A.1`` (page 12) of the submitted manuscript, we have presented a ``simplified version`` of the Open-Det pipeline (Figure ``5``), which highlights the core workflow and provides a clearer illustration of the entire process for better clarity. Additionally, the caption of Figure ``2`` has explicitly indicated the relationships between the two figures, contributing to a better understanding of the main pipeline. We deeply acknowledge the reviewer's suggestion and will further modify Figure ``2`` by reorganizing the figure layout to make it more understandable.
>
> [1] Grounded language-image pre-training. CVPR 20222.
>
> [2] Grounding dino: Marrying dino with grounded pre-training for open-set object detection. ECCV 2024.
>
> [3] Detclipv3: Towards versatile generative open-vocabulary object detection. CVPR 2024.

---

### Decision · Program_Chairs · 2025-05-01

**Decision:**

Accept (poster)

**Comment:**

This paper was reviewed by four experts in the field, receiving mixed evaluations: two Weak Accepts, one Accept, and one Reject. Overall, most reviewers provided positive feedback. The authors have addressed the majority of the concerns raised during the review process. The main remaining issue, raised by reviewer AMke, pertains to citation practices. However, the AC acknowledges that the authors have provided a detailed response to this concern. It is recommended that the authors carefully revise the final version by adding appropriate citations and clearly highlighting the differences from related work. As a result, the final decision is to accept the paper.